# Stimulated plasmon polariton scattering

C. Wolff [1]✉ & N. A. Mortensen [1,2,3]

Plasmon and phonon polaritons of two-dimensional (2D) and van-der-Waals materials have recently gained substantial interest. Unfortunately, they are notoriously hard to observe in linear response because of their strong confinement, low frequency and longitudinal mode symmetry. Here, we propose an approach of harnessing nonlinear resonant scattering that we call stimulated plasmon polariton scattering (SPPS) in analogy to the opto-acoustic stimulated Brillouin scattering (SBS). We show that SPPS allows to excite, amplify and detect 2D plasmon and phonon polaritons all across the THz-range while requiring only optical components in the near-IR or visible range. We present a coupled-mode theory framework for SPPS and based on this find that SPPS power gains exceed the very top gains observed in on-chip SBS by at least an order of magnitude. This opens exciting possibilities to fundamental studies of 2D materials and will help closing the THz gap in spectroscopy and information technology.

[1] Center for Nano Optics, University of Southern Denmark, Campusvej 55, Odense M DK-5230, Denmark. [2] Danish Institute for Advanced Study, University of Southern Denmark, Campusvej 55, Odense M DK-5230, Denmark. [3] Center for Nanostructured Graphene, Technical University of Denmark, Kongens, Lyngby DK-2800, Denmark. ✉email: cwo@mci.sdu.dk

The study of plasmon–polariton excitations in two-dimensional (2D) materials[1] and the related class of van-der-Waals (vdW) materials[2] has recently gained considerable attention since they provide the means to very tightly confine, guide, and manipulate electromagnetic fields from the few-THz range all the way into the mid-infrared. Furthermore and unlike conventional plasmonics based on metals, 2D material plasmon polaritons are highly sensitive to their electromagnetic, electronic, and chemical environment, allowing for great tuning flexibility as well as suggesting their versatile use for sensing[3–5]. As an example, the extreme spatial light confinement paves the way to applications as varied as mid-infrared vibrational fingerprints of proteins[6], control of symmetry-forbidden atomic transitions[7], or single-photon nonlinear optics[8]. However, their greatest strengths—unrivaled mode confinement and operation in a hitherto poorly explored frequency regime—along with their peculiar mode symmetry also turn out to be a nuisance for experimental work. To date, the most promising avenue to overcome the extreme wave number mismatch between polaritons and free-space radiation is by scattering at discontinuities, e. g. introduced by the probe of a scanning near-field optical microscope (SNOM), or material discontinuities designed into the device[9]. As an alternative approach, the generation of graphene plasmon polaritons by difference-frequency generation based on the intrinsic nonlinearities has been studied both theoretically[10,11] and experimentally[12]. This has led to significant insights[13], but is somewhat inefficient and does not seem very practical for applications beyond fundamental research. We suggest that this can be further drastically enhanced and harnessed for practical applications by borrowing ideas from the seemingly unrelated field of opto-mechanics, specifically of Brillouin scattering.

Stimulated Brillouin scattering (SBS) is the inelastic, resonant, and self-amplifying back scattering of light from a propagating acoustic wave in matter[14–16]. From its initial status as an academic curiosity, it has soon proven invaluable to characterize mechanical properties of materials at GHz frequencies—a difficult range for direct mechanical measuring techniques. More recently, it has attracted considerable attention for the realization of flexible yet highly selective optical filters[17], novel light sources[18], the processing and buffering of optical signals[19,20], and for the generation and amplification of coherent hypersonic waves, e.g. following the concept of the so-called phonon laser[21].

We introduce the concept of inelastic, resonant, and self-amplified scattering of light of propagating polaritons especially in 2D and vdW materials; a process that we refer to as stimulated plasmon polariton scattering (SPPS). In analogy to the conceptually similar SBS, this will not only allow for the detection of 2D plasmons at the most convenient optical wavelength, but also to accurately measure both the frequency and damping at the given wave number. In cases where the dispersion relation depends on a parameter (e. g. the Fermi energy $E_F$), the wide-band nature of SPPS allows to characterize this dependency over 1–2 orders of magnitude. In the case of graphene plasmon polaritons (GPPs), this means that all regimes of the dispersion relation (pure intraband scattering, interband corrections, and potentially even nonlocal effects[22–25]) can be experimentally characterized in one setup. Beyond this use in fundamental science, one can expect to harness SPPS for use in optical components, but especially for the tunable narrow-band amplification and optical detection of signals in the THz range. In the remainder of this paper, we describe the principle of SPPS and conclude that it is experimentally observable in a standard silicon slot waveguide covered with an appropriately biased graphene monolayer.

The proposed mechanism bears similarities with the excitation of graphene plasmons[10] through difference-frequency generation

(DFG) via the intrinsic Pockels nonlinearity of graphene[26], but our aim is different. Our focus is not the intrinsic graphene nonlinearity that unlocks DFG, but rather the consequences of this nonlinearity on the interplay between optical and polaritonic fields that are confined in a common long waveguide. We show that this will cause exponential gain for the THz polaritonic wave on a length scale well beyond its linear propagation length provided the nonlinear interaction is sufficiently strong. We find that among the nonlinear processes that can be found or introduced in a system composed of a dielectric waveguide combined with a graphene sheet, the ponderomotive nonlinearity[27] is sufficiently strong, but we stress that it is not necessarily the only possibility.

## Results

**Principle**. The SPPS process is conceptually related to the well-understood SBS process in nanoscale waveguides: in both cases a propagating low-frequency excitation with wave number $q$ modifies the local permittivity of a waveguide through a nonlinear process. In SBS, the excitation is a sound wave, and the nonlinear process is due to photoelasticity and the motion of the dielectric interface. In SPPS, the excitation is a localized polariton and the nonlinear interaction is either due to an intrinsic nonlinearity of the polariton system or to a Pockels nonlinearity in the waveguide. Thus, the polariton creates a traveling low-contrast grating in the waveguide, which can scatter an optical pump wave (with wave number $k$) into a counter-propagating Stokes wave if the difference in optical frequency and momentum matches the polaritonic dispersion relation (phase matching). Assuming that the Stokes shift $\Omega$ is small compared with the optical pump frequency $\omega$, the phase-matching condition is approximately given by the ratio $q \approx 2\,k$ (see Fig. 1). In analogy to SBS, overall conservation of energy and momentum requires that both the polariton field and the optical Stokes wave grow approximately exponentially along the waveguide. This process exists as soon as there is an efficient nonlinear coupling mechanism between the optical pump and the polariton. Naturally, this nonlinear coupling must be strong enough that the process is not immediately quenched by the linear losses experienced by the plasmon–polariton. At least for THz graphene plasmon polaritons, this is actually not as bad as it might perhaps sound. Their propagation loss is in fact very comparable to that of GHz-range acoustic phonons found in SBS as illustrated by the fact that both THz graphene plasmons as well as GHz sound waves in technologically relevant materials[14,16,28–30] have quality factors of orders of magnitude $10^1$–$10^3$. Finally, neither SBS nor SPPS require long-range propagation of their respective non-optical excitations.

**Theory**. We now introduce the basic theoretical framework. In analogy to the theory of SBS[31], we describe the dynamics of the three participating waves within the framework of coupled-mode theory, where we assume a strict slowly varying envelope approximation. This means that the optical pump amplitude $a^{(p)}(y, t)$, the optical Stokes amplitude $a^{(s)}(y, t)$, and the plasmon–polariton amplitude $b(y, t)$ all are assumed to vary slowly on the timescale of the slowest carrier in the system: the polariton frequency. This all leads to the nonlinearly coupled equations

$$\partial_y a^{(p)} + v_a^{-1}\partial_t a^{(p)} + \kappa_a a^{(p)} = -i\omega Q \mathcal{P}_a^{-1} a^{(s)} b^*, \quad (1a)$$

$$\partial_y a^{(s)} - v_a^{-1}\partial_t a^{(s)} - \kappa_a a^{(s)} = -i\omega Q^* \mathcal{P}_a^{-1} a^{(p)} b, \quad (1b)$$

$$\partial_y b + v_b^{-1}\partial_t b + \kappa_b b = -i\Omega Q \mathcal{P}_b^{-1} \left[a^{(p)}\right]^* a^{(s)}, \quad (1c)$$

where we assume the waveguide to extend along the $y$ direction,

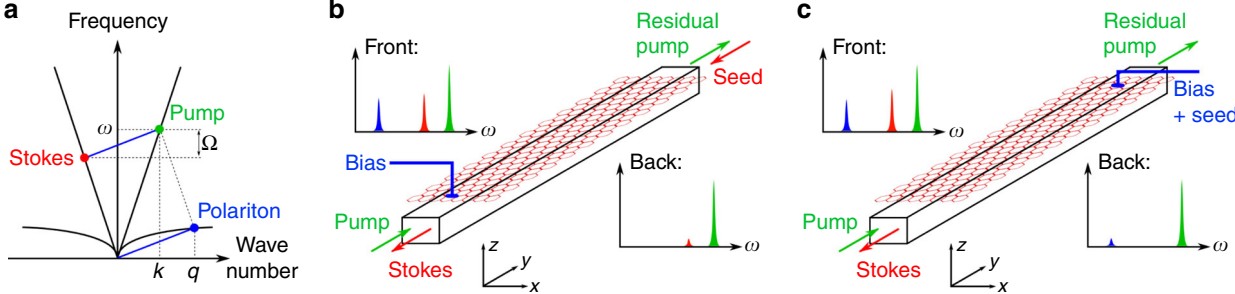

**Fig. 1 Illustration of the phase matching in SPPS and schematics of two potential realizations. a** SPPS interaction in the optical (straight solid) and polaritonic (curved solid) dispersion relation: the pump (green point, frequency $\omega$, and wave number $k$) is scattered into the counter-propagating Stokes mode (red point) where the difference in frequency and momentum (blue line) matches the polaritonic dispersion relation. This polariton mode (blue point, wave number $q \approx 2k$ for angular frequency $\Omega \ll \omega$) is amplified along the waveguide. **b** Conceptual schematic of an SPPS experiment in Stokes-seed configuration. A weak Stokes seed (red arrows) is injected at the rear end, and amplified as it propagates toward the front end of a graphene-covered waveguide. This action is also illustrated by qualitative spectra where the heights of the colored peaks indicate to the relative amplitudes of the matching signals at the front and back of the waveguide. The bias voltage contact and the waveguide geometry shown here were intentionally kept oversimplified at this stage and in reality would require careful engineering. This configuration is best suited for the excitation of polaritons and for characterizing their dispersion relation, which can be tuned via the bias voltage contact. It is also the natural candidate for a first demonstration of SPPS, as all input and output signals are in the optical domain and no injection or detection of THz waves is required. **c** Conceptual schematic of an SPPS experiment in polariton-seed configuration. A weak THz signal is injected through the bias contact (blue), and amplified along the waveguide. Simultaneously, an optical Stokes is generated (illustrated by qualitative spectra similar to panel **b**). This configuration is compelling because of the prospect to amplify or optically detect weak THz signals.

$v_a$ is the group velocity of the optical mode, $\mathcal{P}_a$ is the power to which it has been normalized, and $\kappa_a$ is the optical decay parameter. Their plasmonic counterparts are $v_b$, $\mathcal{P}_b$, and $\kappa_b$, respectively. Natural choices for the normalization powers are $\mathcal{P}_a = \hbar \omega v_a / L$ and $\mathcal{P}_b = \hbar \Omega v_b / L$, respectively, with the unit length of waveguide $L = 1$ m. The interaction is mediated by the nonlinear overlap between the optical pump and Stokes modes, and the permittivity change caused by the polaritonic mode:

$$Q = \left\langle \overrightarrow{e}^{(p)} \middle| \Delta \varepsilon(\overrightarrow{e}^{(pol)}) \middle| \overrightarrow{e}^{(s)} \right\rangle = \int \mathrm{d}^2 A \, \varepsilon_0 \sum_{ij} [e_i^{(p)}]^* e_j^{(s)} \Delta \varepsilon_{ij}(\overrightarrow{e}^{(pol)}),$$

(2)

where the integral is carried out over the cross sectional plane of the waveguide, $\overrightarrow{e}^{(pol)}$ is the electric field distribution of the polariton mode, and $\Delta \varepsilon_{ij}(\overrightarrow{e}^{(pol)})$ includes all suitable second-order nonlinearities in the system. This may include conventional Pockels nonlinearities $\chi^{(2)}$ in the waveguide or the polaritonic material or effects such as the ponderomotive nonlinearity, which we base our example on. In analogy to the theory of SBS[31], we can easily derive from Eqs. ((1a)–(1c)) the stationary power gain

$$G = \frac{2\omega\Omega|Q|^2}{\mathcal{P}_a^2 \mathcal{P}_b \kappa_b},$$

(3)

which is the most interesting quantity in experiments. Besides inconsequential normalization constants and the frequencies $\omega$, $\Omega$, its main ingredients are the nonlinear coupling integral $Q$ and the polaritonic loss parameter $\kappa_b$. Therefore, the reader concerned with loss in graphene might interpret $G$ as a measure for how strong the nonlinear coupling is compared with the linear polaritonic loss. This should be kept in mind when comparing the gain in successful SBS experiments to the numerical values that we calculate further below. The derivation of these equations and those underlying our numerical example is beyond the scope of the main text and provided in Supplementary Note 1.

**Feasibility**. We will now apply this theoretical framework to an illustrative waveguide geometry that has not been optimized for

the strongest possible gain, but rather selected for its simplicity. The main purpose of this example is to demonstrate that the concept is feasible, i.e. that sufficient gain can be achieved in a realistic setting. We focus entirely on the interplay of the opto-plasmonic nonlinearity and linear losses, and leave important technological aspects of an actual experiment out. This includes especially the exact method of doping, which can be achieved, e.g., electrostatically via a metal contact few microns above the structure, with direct electrical contacts[32] or by chemical doping. As a waveguide, we select a slot waveguide composed of two silicon beams with rectangular cross section (dimension $220 \times 275$ nm$^2$) separated by a 150-nm air gap and operated at the standard telecom wavelength 1550 nm corresponding to a photon energy of 0.8 eV. We computed the electric field distribution of the fundamental mode (effective index 1.37) using the commercial finite element software COMSOL (see Fig. 2a). The slot waveguide was selected, because this configuration is known to support waveguide modes that are mostly confined in the gap between the silicon beams and are therefore an excellent starting point for nonlinear photonics in the silicon nanophotonics platform[33]. As a polaritonic system, we select a graphene monolayer that has been sandwiched between thin layers of hexagonal boron nitride to isolate it from adverse substrate effects. The polaritonic damping parameter $\kappa_b$ depends on $E_F$. We calculate it from the Drude relaxation rates (3.5 meV at 300 K and 0.35 meV at 60 K) that were experimentally determined for this type of "vdW-sandwich"[24] and plot it implicitly in Fig. 2b as the polaritonic quality factor $\Omega/(2 v_b \kappa_b)$. As mentioned before, this quality factor is the appropriate measure when comparing with loss found in SBS and other related scattering processes.

We consider two possible placements of the graphene: either as a narrow ribbon inside the waveguide gap for maximal field enhancement or placed on top of the waveguide as a more practical arrangement. We note that the dispersion relation of such ribbons differs from that of infinitely extended graphene due to the lateral confinement[34]. In our example system, the second-order nonlinearity stems entirely from the intrinsic ponderomotive force of the electron system in graphene. It is a direct consequence of the dependence of the optical properties on the

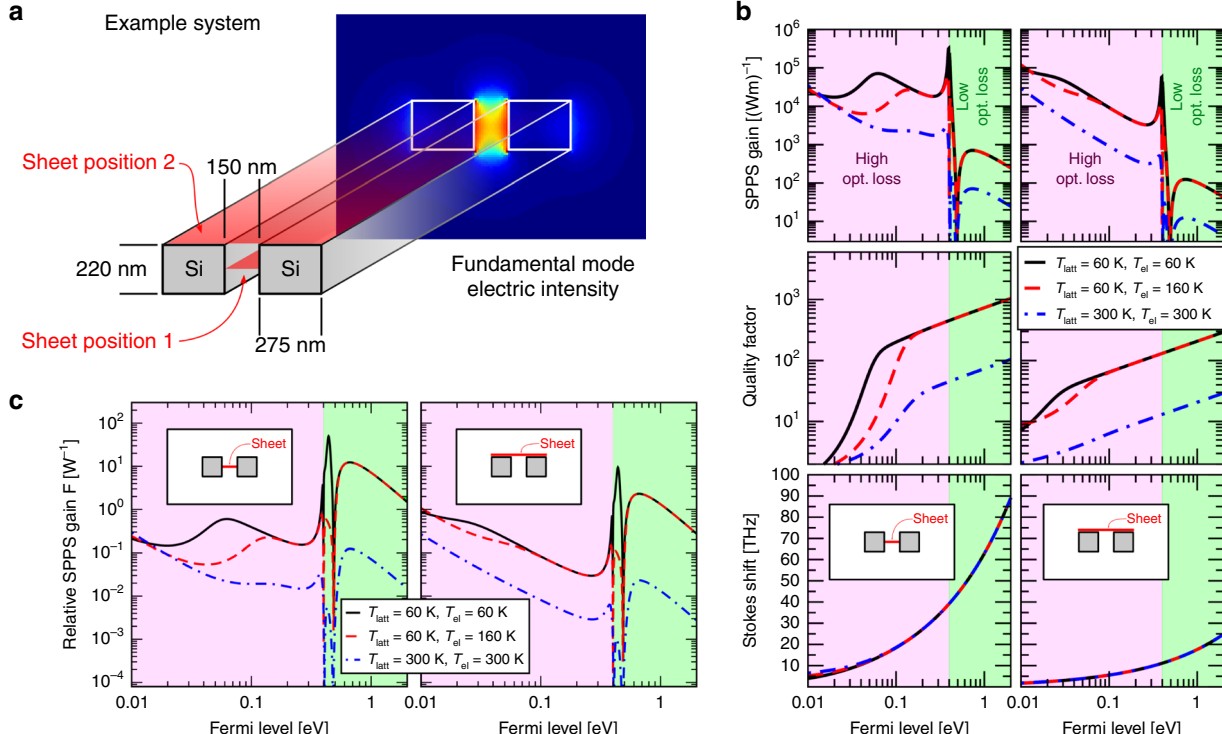

**Fig. 2 Numerical feasibility study of a simple SPPS experiment based on a silicon slot waveguide and a graphene ribbon. a** Sketch of the example waveguide and electric intensity distribution of the fundamental mode at 1550 nm. The red planes indicate the two considered placements of the graphene sheet. **b** Total SPPS power gain, quality factor (i.e., polaritonic loss), and Stokes shift (i.e., polaritonic frequency) of the two example structures and at three different combinations of temperatures as functions of the Fermi level $E_F$. The temperatures are: $T = 300$ K, thermal equilibrium between phonon and electron gases (blue dash-dotted); $T = 60$ K, thermal equilibrium (black solid); red dashed: lattice at $T_{latt} = 60$ K and electron gas at $T_{el} = 160$ K (red dashed) as a result of high pump powers (see "Discussion" and Supplementary Note 2). The pink- and green-shaded areas indicate whether the optical pump can or cannot drive interband transitions, leading to a high-loss and a low-loss regime for the pump and Stokes signals. At the transition, the resonant interband contribution to the ponderomotive force creates a pronounced peak in the gain. **c** Relative SPPS-gain figure $F = G/\kappa_a$ as a function of the Fermi level.

Fermi level, and has been studied based on a pure intraband model for the optical conductivity[35]. In addition, we also consider the interband contributions and finite-temperature corrections to the ponderomotive force that we have published elsewhere[27]. We should stress that while the ponderomotive interaction constitutes a second-order nonlinearity, it must not be confused with the second-order nonlinear electric susceptibility $\chi^{(2)}$. The latter is a third-rank tensor relating three electric field vectors, and the former is the sensitivity of the conductance tensor to variations in the scalar carrier density and therefore in general described by only a second-rank tensor.

In Fig. 2b, we show the calculated SPPS gain, quality factor, and Stokes shifts of our example system for either sheet placement (left column: inside the gap, right column: on top of the waveguide) and at a moderately low temperature of 60 K as well as at room temperature. We find Stokes shifts throughout the entire THz range, and for low Fermi levels we predict SPPS gains in excess of $10^4$ (Wm)$^{-1}$ over a fairly large parameter range with peaks approaching $10^6$ (Wm)$^{-1}$. This has to be compared to 1–10 (Wm)$^{-1}$ for SBS in optical fibers[36,37], 100–1000 (Wm)$^{-1}$ in chalcogenide rib waveguides[16,28] and 1000–10,000 (Wm)$^{-1}$ in well-engineered silicon waveguides[29,30,38]. This demonstrates that SPPS can be expected to provide levels of gain that are very competitive with similar nonlinear processes, such as SBS, even though the graphene sheet introduces linear loss, which can limit the useful waveguide length to below 1 mm. To avoid confusion, we note that SBS-gains for bulk materials are usually specified in

alternate units of m/W and can only be compared with our number when divided by a suitable mode area.

## Discussion

The example system of Fig. 2 was selected for its simplicity, maintaining conservative numbers for the aspect ratios, and feature and gap sizes. Besides the technical problem of transferring a graphene sandwich, the main experimental challenge is the linear loss introduced by the graphene. For a given photon energy (e. g. 0.8 eV as chosen in our example) and variable Fermi level, this introduces two quite distinct regimes: If the Fermi level is below half the photon energy (here: 0.4 eV), graphene is extremely lossy due to interband transitions, leading to high linear waveguide loss (in our example reaching power loss values up to 10,000 dB/cm). For Fermi levels above half the photon energy, however, the loss drops by several orders of magnitude (we calculate values as low as 1 dB/cm at 60 K and just above the interband threshold). Since the linear optical loss limits the useful waveguide length, a natural figure of merit is the ratio between gain and loss:

$$F = G/\kappa_a, \qquad (4)$$

which takes units of W$^{-1}$ and whose inverse $1/F$ is the pump power level necessary to achieve net gain (the effect can remain detectable at considerably lower pump levels, e.g., by pump modulation and lock-in amplification of the output Stokes). We plot this in Fig. 2c. For most Fermi levels, we find values around

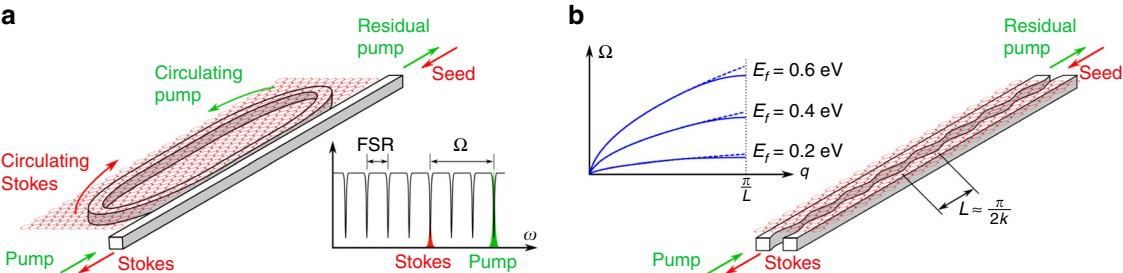

**Fig. 3 Sketches of two possibilities to boost the SPPS response via polaritonic dispersion engineering. a** A ring topology for the SPPS-active element side-coupled to a bus waveguide increases the gain dramatically, but requires the Stokes shift Ω to align with a multiple of the free spectral range (FSR) of the ring, significantly restricting any tunability. **b** A corrugation at 1/4 the pump wavelength in the waveguide introduces a "slow-light'' regime with the corresponding Purcell enhancement for the polariton mode (solid blue lines compared with dashed for the uncorrugated waveguide). As the enhancement appears at a fixed wave number, the polaritonic Purcell enhancement is controlled by the pump frequency and can be achieved virtually irrespective of the actual Stokes shift Ω, which remains tunable, e.g., via the Fermi level.

$0.1\ldots1\,\text{W}^{-1}$, which is enhanced up to $100\,\text{W}^{-1}$ due to a ponderomotive resonance at the interband threshold. This means that peak pump powers from 10 W down to 10 mW would provide net gain (the latter at liquid nitrogen temperatures and for waveguide lengths in the low millimeter range). This is very feasible with state-of-the-art silicon photonics techniques, and sub-ns pulsed light sources would prevent damage due to excessive dissipation in the graphene and free carrier absorption in silicon. We note that for pump powers on the higher side, pump dissipation in the graphene causes a relevant difference between the temperatures of the electron gas and the phonon gas. This leads to reduced SPPS gain and increased optical loss while leaving the polaritonic quality factor for the most part unchanged. The exact threshold for this effect depends strongly on optical pump power, (lattice) temperature, Fermi level, and waveguide design (see the Supplementary Note 2 for further discussion). For our configuration "1" at 60 K with $E_F \geq 0.4\,\text{eV}$, we estimate a temperature difference of 10 K at 300 mW pump power. To illustrate the effect, we show in Fig. 2 the impact of a very strong imbalance of 100 K expected for 3 W pump at 60 K. We note that this power level is already close to the destruction threshold of a silicon waveguide even with short picosecond pulses, so we present an extreme case. Finally, we find that the overall performance declines above $E_F \approx 0.5\,\text{eV}$, despite a continuously increasing plasmonic quality factor. This is due to the resonant ponderomotive interaction and means that doping beyond the currently feasible levels would not provide any further benefit.

We will now present some guidelines for the design of a more sophisticated geometry with superior performance over our simple slot-waveguide example. First, we point out that the scaling between the polariton wave number $q$ and the quality factor $\Omega/(2\nu_b\kappa_b)$ is counterintuitive to people from the optomechanics community. In SBS, acoustic loss grows superlinearly as $q$ is increased, leading to a decrease in the quality factor. As a result, SBS-active elements are ideally designed to have a low optical mode index and a high acoustic mode index. Curiously, we find the opposite effect in SPPS: the polaritonic quality factor increases as the wave number and frequency are increased. We believe this to be a direct result of decreasing polaritonic mode confinement. As a result, high-mode-index optical waveguides are more viable, and especially the use of plasmonic waveguides based on noble metals appears quite interesting. Even the small propagation lengths are not necessarily a problem, because of the high gain and the inherent high optical loss of materials such as graphene in the interband regime. Furthermore, we point out that any form of field enhancement will improve the relative gain figure $F$. While it is

true that placing the polaritonic sheet in higher field enhancements inevitably increases the linear loss due to interband transitions, the SPPS process (like SBS and Raman scattering) is effectively a third-order nonlinearity and thus is bound to over-compensate the increase in loss.

Second, we find that dielectric loading of the polaritonic mode is quite beneficial. This is the reason why a graphene sheet positioned above a slot waveguide provides gains that are comparable to those of a narrow ribbon squeezed inside the gap, even though it is subject to considerably lower intensity enhancement and overall poorer mode overlap. It is not clear how screening of the polaritonic mode (e.g., due to the use of gold-plasmonic waveguides) affects the overall gain.

Finally, we anticipate that the gain can be dramatically boosted by dispersion engineering of both the optical and the polaritonic mode. One example for such a system is to shape the SPPS-active element into a ring that is side-coupled to a bus waveguide (Fig. 3a). This topology has been studied thoroughly in the context of SBS and can dramatically increase the overall gain, even allowing for spontaneous oscillation (lasing)[39]. However, it requires careful choice of the coupling parameter and close matching between the Stokes shift and the ring's free spectral range. As a result, it is a well understood and easily implementable concept, which, however, negates the opportunities offered by tuning the Fermi level. A second possibility is to corrugate the waveguide to introduce a band edge with associated slow-light regime. This is known to enhance SBS in slow-light fibers[40]. Alternatively, it is also possible to create a polaritonic band edge (Fig. 3b), reducing the polaritonic group velocity and hence mode power $\mathcal{P}_b$ that appears in the denominator of Eq. (3). Despite the resonant nature of such a "slow-light" regime, this Purcell enhancement would be in fact broadband. This may seem counterintuitive at first, but is just a result of the fact that the band edge is always positioned at the Brillouin zone border, i.e. for a fixed wave number. Since $q \approx 2k$ is effectively fixed by the optical pump wave and Ω follows, e.g., as a function of the Fermi level, a corrugation with a period of 1/4 of the pump wavelength in the waveguide will always introduce a Purcell enhancement irrespective of the value of Ω.

Beyond the value of SPPS as a tool for the fundamental science of atomically thin materials, we also anticipate potential practical applications once geometries with optimized gain have been developed. The first, obvious possibility is to adapt some of the current applications of SBS such as integrated light sources (similar to Raman lasers), optical signal filtering, and especially sensing, which is one of the areas where SBS is currently commercially applied. Like its acoustic counterpart, the polaritonic dispersion relation is highly sensitive to variations in the

surrounding material. While the current SBS sensors mainly detect variations of the cladding's acoustic impedance and static strain fields, similar SPPS sensors involving a sheet of 2D material would be highly sensitive to very thin adsorbed layers either through a change of permittivity, a reduction in the carrier mobility or shifts in the Fermi level[4]. Furthermore, the principles of Brillouin optical correlation domain analysis (BOCDA) could be adapted to pinpoint the perturbation along an SPPS sensor with sub-mm resolution[41].

Finally, we would like to emphasize another aspect that makes SPPS interesting for practical applications: the amplification and detection of signals in a narrow-frequency band that can be selected anywhere in the THz- to mid-IR band. Like SBS, SPPS is a self-amplifying process that increases the amplitude of both the low-frequency excitation (sound in the case of SBS, polaritons in SPPS) as well as the optical Stokes signal exponentially along the wave-guide. This means that a weak THz signal injected into a graphene sheet at the back of an SPPS-active waveguide could be detected in the optical domain as the Stokes signal. In addition, the amplified THz signal could be picked off at the front of the waveguide. Both the detection and amplification would be restricted to the narrow-frequency window given by the plasmon frequency and quality factor at the given wave number, and could be tuned within a wide range via the Fermi level. Such a tunable narrow-band detector in conjunction with a wide-band source (e.g., a thermal emitter) would have the same versatility as a narrow-band source in conjunction with a broadband detector, and could be of considerable value to spectroscopy in the far-IR and THz regime.

In summary, we described the optical coupling to (e. g. plasmonic) polaritons in 2D and vdW materials through a new physical process that we call stimulated plasmon–polariton scattering. We outlined the theoretical framework, and based on experimentally verified material parameters we showed that the process can be observed in an experimentally straight-forward system at moderate temperatures despite significant optical loss and that even net gain for a Stokes signal can be achieved with a continuous-wave pump in this example at the appropriate Fermi level.

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

## Acknowledgements

C.W. acknowledges funding from a MULTIPLY fellowship under the Marie Skłodowska-Curie COFUND Action (grant agreement no. 713694). The Center for Nano Optics is financially supported by the University of Southern Denmark (SDU 2020 funding). N.A.M. is a VILLUM Investigator supported by VILLUM Fonden (grant no. 16498).

The Center for Nanostructured Graphene is sponsored by the Danish National Research Foundation (Project No. DNRF103). We are deeply indebted to P.A.D. Gonçalves, Joel Cox, and especially C. Tserkezis for valuable comments on the paper.

## Author contributions

C.W. conceived and conducted the research. C.W. and N.A.M. discussed the results and contributed to the paper.

## Competing interests

The authors declare no competing interests.
