## [Peer Review File · Nature Communications]

Reviewers' comments:

Reviewer #1 (Remarks to the Author):

Taking advantage of the concept of Brillouin scattering in opto-mechanics, Wolff and Mortensen introduce an analog in the context of 2D polaritons. They dub it "stimulated plasmon polariton scattering" (SPPS) and present the principle of this mechanism, which is expected to be useful in measuring dispersion and lifetimes of polaritons in 2D materials in a metallic-tip-free setup, but also for a variety of applications.

The paper is well written, figures are well described. The results are fully numerical (COMSOL): I therefore have no way to check whether they are right or wrong by myself, but they seem to be sound and free of macroscopic mistakes.

I found some typos (see for example definition of Fermi energy, once called E_F and another time called E_f , etc) here and there. Please proofread more carefully.

The idea is intriguing and may really work out provided that certain experimental difficulties in fabricating devices are surpassed. I believe that this work should be published in Nature Communications provided that the authors take care of the following two concerns:

1) As a theorist, I'd like to see a small Supplementary Information file, where a brief derivation of the coupled nonlinear equations (1a)-(1c) is presented. I understand the authors want to postpone the presentation of a detailed derivation to a long technical paper, as they state in page 3, left column, but I find this a bit odd.

A small Supplementary File of one page, say, will not harm their long technical paper and will be useful to the readers of the present Letter, who are curious about the assumptions and approximations that lead to Eqs. (1a)-(1c).

2) As discussed by the authors, nonlinearities are crucial on the physics of SPPS, and are parametrized via the nonlinear second-order susceptibility tensor in Eq. (2).

In this respect, I have two remarks.

2a) One is merely bibliographic.

Ways of launching and detecting graphene plasmons, which do not rely on SNOM, have been recently discussed in the following papers, which should be cited:

- X. Yao, M. Tokman, and A. Belyanin, Phys. Rev. Lett. 112, 055501 (2014)
- T. J. Constant, S. M. Hornett, D. E. Chang, and E. Hendry, Nat. Phys. 12, 124 (2016)
- H. Rostami, M.I. Katsnelson, and M. Polini, Phys. Rev. B 95, 035416 (2017)

These rely also on a second-order nonlinear process intrinsic to graphene and parametrized via a nonlinear second-order susceptibility tensor.

Since the plasmon carries a finite in-plane momentum q , graphene's inversion symmetry is effectively broken by q

and this leads to a finite second-order susceptibility tensor.

2b) The second remark is merely a curiosity.

I was wondering whether the authors have included such intrinsic nonlinearities in their nonlinear second-order susceptibility tensor, or whether that would lead to "double counting" and therefore should not be included in the right-hand side of Eq. (2).

This is just a personal curiosity, which I can't answer by myself, because I have not seen a derivation of Eqs. (1a)-(1c).

No changes are requested in the revised manuscript in this second respect.

In summary, I strongly believe that the present manuscript should be published immediately in Nature Communications, provided that the authors have taken care of 1) and 2a) above.

Reviewer #2 (Remarks to the Author):

In their manuscript entitled "Stimulated plasmon polariton scattering" the authors C. Wolff and N. A. Mortensen draw an analogy between plasmons and acoustic phonons as they propose a new strategy for enhancing and tailoring plasmon-polariton nonlinearities in guided-wave systems. I'm grateful that the authors do not sweep any of the nontrivial aspects of the system aside. However, the experimental feasibility of proposed interactions come with numerous challenges. That said, the authors are putting forward an interesting proposal, and to my knowledge, the ideas proposed are new.

For these reasons I am not opposed further consideration by nature communications provided that they address several queries below:

1) Following equation 3, the authors state that " the derivation and detailed discussion of these equations is beyond the scope of this letter will be published separately". It seems to me that this derivation and discussion are essential to this paper. I understand the desire to keep this publication short, on the other hand, I think it essential to include this content in a supplement. Otherwise, the paper seems incomplete.

2) As for the experimental relevance of this proposal, it is unclear that slot waveguides of the type proposed offer a viable experimental solution. To date, slot waveguide systems suffer from extraordinarily high linear losses, because the same confinement mechanism also produces strong overlap with roughness and surface imperfections that degrade the optical performance of these waveguides.

3) A nontrivial aspect of this experimental proposal that has not been addressed is heat dissipation. If this coupling mechanism is feasible, it is unclear whether or not lose 2D materials will remain stable as rather large amounts of optical power are absorbed into atomically thick layers. It seems that a rudimentary thermal model is necessary to address this problem

Some minor comments and corrections follow:

In figure 2 the authors state that the "pink and green shaded areas indicate whether the pump can or cannot drive independent transitions". The statement is ambiguous please clarify.

The authors state that "dielectric loading of the polaritonic mode is quite beneficial" on page 5. The intended meaning is unclear.

Reviewer #3 (Remarks to the Author):

I do not recommend this manuscript for publication since it contains neither experimental confirmation that the proposed by authors device does work, nor a careful, detailed, comprehensive and – most importantly – verifiable theory. The authors reports that the development of a detailed theory "is beyond the scope of this Letter" but then it does not make sense to publish this Letter.

Comments of Referee 1 (italic font) and our responses

Referee:

Taking advantage of the concept of Brillouin scattering in optomechanics, Wolff and Mortensen introduce an analog in the context of 2D polaritons. They dub it "stimulated plasmon polariton scattering" (SPPS) and present the principle of this mechanism, which is expected to be useful in measuring dispersion and lifetimes of polaritons in 2D materials in a metallic-tip-free setup, but also for a variety of applications.

The paper is well written, figures are well described. The results are fully numerical (COMSOL): I therefore have no way to check whether they are right or wrong by myself, but they seem to be sound and free of macroscopic mistakes. I found some typos (see for example definition of Fermi energy, once called E_F and another time called E_f , etc) here and there. Please proofread more carefully.

The idea is intriguing and may really work out provided that certain experimental difficulties in fabricating devices are surpassed. I believe that this work should be published in Nature Communications provided that the authors take care of the following two concerns:

- 1. As a theorist, I'd like to see a small Supplementary Information file, where a brief derivation of the coupled nonlinear equations (1a)-(1c) is presented. I understand the authors want to postpone the presentation of a detailed derivation to a long technical paper, as they state in page 3, left column, but I find this a bit odd. A small Supplementary File of one page, say, will not harm their long technical paper and will be useful to the readers of the present Letter, who are curious about the assumptions and approximations that lead to Eqs. (1a)-(1c).*
- 2. As discussed by the authors, nonlinearities are crucial in the physics of SPPS, and are parametrized via the nonlinear second-order susceptibility tensor in Eq. (2).*

In this respect, I have two remarks.

One is merely bibliographic. Ways of launching and detecting graphene plasmons, which do not rely on SNOM, have been recently discussed in the following papers, which should be cited:

- X. Yao, M. Tokman, and A. Belyanin, Phys. Rev. Lett. 112, 055501 (2014) - T. J. Constant, S. M. Hornett, D. E. Chang, and E. Hendry, Nat. Phys. 12, 124 (2016) - H. Rostami, M.I. Katsnelson, and M. Polini, Phys. Rev. B 95, 035416 (2017)

These rely also on a second-order nonlinear process intrinsic to graphene and parametrized via a nonlinear second-order susceptibility tensor. Since the plasmon carries a finite in-plane momentum q , graphene's inversion symmetry is effectively broken by q and this leads to a finite second-order susceptibility tensor.

- 3. The second remark is merely a curiosity. I was wondering whether the authors have included such intrinsic nonlinearities*

in their nonlinear second-order susceptibility tensor, or whether that would lead to "double counting" and therefore should not be included in the right-hand side of Eq. (2). This is just a personal curiosity, which I can't answer by myself, because I have not seen a derivation of Eqs. (1a)-(1c). No changes are requested in the revised manuscript in this second respect.

In summary, I strongly believe that the present manuscript should be published immediately in Nature Communications, provided that the authors have taken care of 1 and 2a above.

Response:

Our response to the points raised by the reviewer:

We thank the reviewer for their very positive assessment of our work. We respond to the queries as follows:

- 1. This point was raised by all referees and in response we added a supplementary note containing the underlying theory. We did not restrict ourselves to a single-page summary, but rather attached all relevant sections of the planned separate theory paper, which will no longer be published as originally intended.**
- 2. We are very thankful to the reviewer for pointing us towards these works, which we added to the discussion of the scientific context of our work in the introduction.**
- 3. As for the question of intrinsic nonlinearities, they were indeed included in the calculation are in fact the dominant contribution to the overall nonlinear coupling. Even though the reviewer explicitly did not ask for it, we included some detailed information on the intrinsic nonlinear processes in graphene to the supplementary material. In short: There are at least two types of intrinsic second-order nonlinearities in graphene. One is the ponderomotive force, which is due the fact that an optical intensity gradient breaks inversion symmetry. This is in fact what drives the interaction in our paper. A second intrinsic effect is the Kerr-like nonlinearity induced by the fact that the wave number of a homogeneous oblique illumination breaks inversion symmetry. This is what the reviewer most likely has in mind and some theory can be found e.g. in [Scientific Reports 7, 43843 (2017)]. We did not include this effect, mainly because we expect it to be small in this setting due to the very similar but opposite wave numbers of the optical pump and Stokes. However, the reviewer is correct that it would be a worthwhile topic to investigate and we can state that it would not be a case of double-counting as far as we are aware.**

Comments of Referee 2 (*italic font*) and our responses

Referee:

In their manuscript entitled "Stimulated plasmon polariton scattering" the authors C. Wolff and N. A. Mortensen draw an analogy between plasmons and acoustic phonons as they propose a new strategy for enhancing and tailoring plasmon-polariton nonlinearities in guided-wave systems. I'm grateful that the authors do not sweep any of the nontrivial aspects of the system aside. However, the experimental feasibility of proposed interactions come with numerous challenges. That said, the authors are putting forward an interesting proposal, and to my knowledge, the ideas proposed are new.

For these reasons I am not opposed further consideration by nature communications provided that they address several queries below:

- 1. Following equation 3, the authors state that " the derivation and detailed discussion of these equations is beyond the scope of this letter will be published separately". It seems to me that this derivation and discussion are essential to this paper. I understand the desire to keep this publication short, on the other hand, I think it essential to include this content in a supplement. Otherwise, the paper seems incomplete.*
- 2. As for the experimental relevance of this proposal, it is unclear that slot waveguides of the type proposed offer a viable experimental solution. To date, slot waveguide systems suffer from extraordinarily high linear losses, because the same confinement mechanism also produces strong overlap with roughness and surface imperfections that degrade the optical performance of these waveguides.*
- 3. A nontrivial aspect of this experimental proposal that has not been addressed is heat dissipation. If this coupling mechanism is feasible, it is unclear whether or not lose 2D materials will remain stable as rather large amounts of optical power are absorbed into atomically thick layers. It seems that a rudimentary thermal model is necessary to address this problem*

Some minor comments and corrections follow:

- In figure 2 the authors state that the "pink and green shaded areas indicate whether the pump can or cannot drive independent transitions". The statement is ambiguous please clarify.*
- The authors state that "dielectric loading of the polaritonic mode is quite beneficial" on page 5. The intended meaning is unclear.*

Response:

Our response to the points raised by the reviewer:

- 1. In response to this point (and similar points raised by the other**

two reviewers) we added a somewhat detailed derivation of the equations shown in the manuscript and those that underlie our numerical results. It can be found as a supplementary note.

2. We thank the reviewer for pointing us towards the problem of surface roughness. It is certainly true that slot waveguides are more prone to this effect. However, as mentioned in the main text, we anticipate waveguide lengths of the order of millimeters. For such relatively short structures, a loss of 1-5 dB/cm as reported e.g. in [Optics Express 19, 26275-26282 (2011)] should not be prohibitive. In fact, we find the general trend that when field concentration increases the SPPS-gain, then this will usually over-compensate increased loss, because the loss scales roughly linearly with intensity whereas the gain grows quadratically. We would like to stress that the waveguide design presented here is meant as a relatively simple proof-of-principle proposal for a new effect and we fully agree that considerable engineering would be required to find the best balance between loss and gain in more application-oriented structures.
3. The reviewer is correct, dissipation is clearly a problem in many nonlinear experiments. In fact, our main concern is not the linear loss in the graphene sheet (graphene appears to be rather stable, especially when sandwiched in boron nitride). Instead, we would be more concerned about free carrier absorption due to two-photon absorption in the silicon. However, we are confident that both problems can be circumvented using a pulsed light source. A light pulse of 1-10W peak power and 1ns duration at a kHz-range repetition rate would be quasi-stationary for the purpose of the SPPS-process while faster than the free carrier lifetime and would deposit < 1 mW of average power across the full length of waveguide while staying below the single-shot damage threshold.

The reviewer is correct that we should have addressed this in our paper. In the main text we did mention pulsed light sources as a means to reach sufficiently high pump powers without risking destruction of the device. We rephrased the paragraph to make this point more explicit.

Regarding the minor points of criticism:

- We are sorry, but we cannot find the word “independent” in the caption. In our files the caption states “pink and green shaded areas indicate whether the pump can or cannot drive inter-band transitions.” Anyway, we ensured that the re-submitted document contains the correct word.
- Dielectric loading refers to the practice of increasing the effective index of a plasmonic waveguide (be it a metal or graphene) by covering it with a high-index dielectric.

Comments of Referee 3 (italic font) and our responses

Referee:

I do not recommend this manuscript for publication since it contains neither experimental confirmation that the proposed by authors device does work, nor a careful, detailed, comprehensive and - most importantly - verifiable theory. The authors reports that the development of a detailed theory "is beyond the scope of this Letter" but then it does not make sense to publish this Letter.

Response:

Our response to the points raised by the reviewer:

We take the reviewer's point. We maintain that a derivation of the theory is beyond the scope of the main text, and therefore added it as a supplementary note.

We are confident that this supplementary will provide enough ground for the reviewer to make a professional assessment regarding the correctness of our findings.

We note that the reviewer criticised only the lack of information to verify the correctness of our findings. They did not criticise our work with respect to impact or novelty, which would have been possible with the information given in our initial submission.

Decision on
manuscript
NCOMMS-19-
04521A

Message:

Dear Dr Wolff,

Please allow me to apologize for the delay in getting a decision to you. The reviewers needed more time to submit their report due to busy schedules. Your manuscript entitled "Stimulated plasmon polariton scattering" has now been seen by 2 referees, whose comments are appended below. In the light of their collective advice I regret to inform you that we cannot publish your manuscript in Nature Communications.

You will see that reviewer #3 raises quite substantial concerns regarding feasibility of the observation. While I realize this is a theoretical work, it is clear that we cannot proceed without substantial work detailing how this phenomenon could be observed and unfortunately, these reservations are sufficiently important to preclude publication of this study in Nature Communications. We have also written to reviewer #2 in confidence regarding the concerns of #3, and they have agreed that these concerns are important. Given the collective feedback, we feel it is best at this stage for you to seek publication elsewhere.

If you opted into the journal hosting details of a preprint version of your manuscript via a link on our dedicated website (<https://nature-research-under-consideration.nature.com>), please note that we will now remove these details as your manuscript is no longer under consideration at Nature Communications. For more information, please refer to our FAQ page at <https://nature-research-under-consideration.nature.com/posts/19641-frequently-asked-questions>

I am so sorry that we cannot be more positive on this occasion and thank you for the opportunity to consider your work.

Best regards,
Lina

Dr Lina Persechini
Associate Editor & Team Manager
Nature Communications
Lina.Persechini@nature.com

Reviewers' comments:

Reviewer #2 (Remarks to the Author):

I'm satisfied with the changes that the authors have made to address the queries posed. In particular, I'm grateful that the authors have now included in-depth supplementary information. The supplementary information answers numerous technical questions and addresses all of my prior concerns regarding the reproducibility of their results.

With this additional information and the revisions that they have made the manuscript, I feel strongly that this paper should be accepted in nature communications. This manuscript explores some intriguing new ideas, and I think it will definitely spur a great deal of interest at the intersection between nonlinear optics and two-dimensional electronic materials.

Reviewer #3 (Remarks to the Author):

The authors put forward an idea of „stimulated plasmon polariton scattering“

(SPPS), to use in 2D materials like graphene, for studying fundamental physics and for some applications. The idea of SPPS is similar to the stimulated Brillouin scattering (SBS) phenomenon known in nonlinear optics. The authors claim that SPPS power gain in the proposed experiments and structures can be sufficiently large under achievable experimental conditions. The paper is theoretical.

In my opinion, it is quite possible that the idea is interesting and the proposed effects may work indeed. But I have not found in the paper clear evidences that the observation of the proposed effects is realistic.

There exists a very large difference between the SBS and the proposed SPPS phenomena. The propagation length of sound waves in liquids and solids can be as large as kilometers, while the propagation length of 2D plasmons in 2D materials is as small as just a few microns. Therefore the authors had to clearly show that the nonlinearity of the plasmon-light interaction is so large that it can overcome the strong damping of plasma waves in the system. This is the most important point and this had to be done with all details to convince the reader that the idea is realistic indeed. I have not found details in the authors' arguments and their conclusions are not convincing at all.

Let me illustrate said above by some examples.

1. In Figure 1 the authors show schematic of the SPPS experiment: a rectangular dielectric optical waveguide is covered by a graphene sheet and the nonlinear plasmon-light interaction should lead to the desired effect. The general idea is clear but there are some questions to its practical realization. In particular, we read "A weak THz signal is injected through the bias contact (blue) and amplified along the waveguide". How the THz wave propagating in free space (wavelength \sim hundred of μm) will be injected into the waveguide with dimensions $\sim 200\text{ nm}$? I believe that a lot of energy will be lost on the interface between the tens-of- μm THz antenna and sub- μm optical waveguides. This issue is not discussed in the paper.

2. In Figure 2(a) the authors suggest two possible realizations of the device. Now we see two dielectric waveguides instead of one from Figure 1. Why two waveguides are better than one? In my opinion, if the graphene sheet lies on the waveguide the light-plasmon interaction (the integral in Eq. (2)) will be larger than when a piece of graphene hangs over the gap of 150 nm width (sheet position 2); this hanging piece of graphene will contribute to the plasmon damping and will work against the desired effect. The sheet position 1 causes even more questions. The optical field is mainly concentrated inside the waveguides and substantially decreases in the gap between the waveguides which is seen from the authors' CONSOL calculations. Placing graphene between the two waveguides will reduce the interaction again as compared to the structure in Figure 1. Why the authors chose the two-waveguide structure is not discussed in the paper. If my intuitive arguments are wrong and the authors did perform calculations of the field distributions in and around the waveguides and found the best positions for the graphene sheets indeed, why these calculations are not shown and not discussed in detail in the paper?

3. It seems that the difference between positions 1 and 2 is in the dielectric constant of the environment, $\epsilon=1$ in the position 1 and $\epsilon=(1+\epsilon_{\text{substrate}})/2$ in the position 2: due to this difference the plasmon frequency and the quality factor are higher in the position 1 than in the position 2 (unfortunately I have to guess here since the authors did not explain this point). However in typical experiments graphene lies on a substrate. If the authors imply just freely hanging graphene in the position 1 I am wondering how this can be practically realized in a real device? If in the position 1 graphene lies on a substrate than the advantage of this geometry ($\epsilon=1$) disappears.

4. In Figure 2(a) the waveguide material is assumed to be Si. But Si is a semiconductor often doped by some amount of impurities. This will lead to an additional damping of 2D plasmons and maybe to a damping of the optical modes.

The propagation of optical modes in the Si waveguides is also not explained. The only information that the effective index equals 1.37 is insufficient; moreover this number seems to be too low for Si waveguide (the dielectric constant of Si is 11.7). Usually experimentalists use dielectric materials for the waveguides like silicon oxide or silicon nitride. Why semiconducting silicon is supposed to be used for SPPS in this work?

5. The authors assume to change the Fermi level from ~ 0.01 eV up to ~ 2 eV (Figure 2(b)). But, firstly, it is unrealistic to push the Fermi level in graphene above ~ 0.4 eV, therefore the very high quality factors and very broad frequency ranges "achieved" at Fermi energies up to ~ 2 eV should be reduced by several times or even by an order of magnitude in realistic structures. Secondly, changing the Fermi level assumes that graphene lies on a SiO₂/doped Si substrate and E_F is varied by the gate voltage applied between graphene and Si. But then, if the hanging graphene is assumed to be in the sheet position 1, all left plots drawn in Figures 2(b,c) are useless.

6. If graphene lies on a SiO₂/doped Si substrate (in order to control E_F) what is the 2D plasmon spectrum in this paper? As far as I understand (I have to guess again since the authors do not discuss this point) the authors assume the square-root dispersion of 2D plasmons, $\omega^2 \sim q$; this can also be seen from the sketch in Figure 1(a). But in the graphene-dielectric-metal (or heavily doped semiconductor) system the spectrum of 2D plasmons is linear, $\omega^2 \sim q^2 d$ at $qd < 1$, where d is the dielectric thickness. If to take this fact into account the plasmon frequency and the quality factor will be further reduced. This important practical issue is not discussed in the paper.

7. The authors give minimal information about how the frequency of 2D plasmons (Stokes shift) and their propagation length (quality factor) are plotted in Figure 2(b). No one formula is given here. They write "The polaritonic damping parameter κ_b depends on E_F . We calculate it from a Drude relaxation rate that was experimentally determined for this type of "vdW-sandwich" [24] and plot it implicitly in Fig. 2b as the polaritonic quality factor $= \Omega / (2v_b \kappa_b)$ ". Why the authors did not give here just two numbers, the relaxation rates for 60 K and 300 K, which they extracted from [24]? Why should I or other readers read the long paper [24] only in order to find numbers which the authors used in their paper? I have found the number of 1.6 ps for graphene at 60 K and could get a figure similar (but not identical, again due to the lack of information) to that shown in Figure 2(b) but I still have no idea how the authors plotted the curves for 300 K. Why should the readers guess what the authors' thoughts were and not just read what they really did?

8. It is unclear at all how the authors calculated the SPPS gain (2) and plotted the corresponding Figures 2(b,c). The value Q in (2) depends on the overlap integral and the nonlinear susceptibility $\chi^{(2)}$. In the SBS effect the interaction takes place in the whole (3D) volume of the waveguide. In the considered here SPPS effect the integration goes over a much smaller volume (over just a 2D plane if $\chi^{(2)}$ is due to the nonlinearity of graphene). Here the authors had to show that, in spite of the thin interaction area the resulting gain is sufficient to overcome losses. But no results of such analysis are shown. Then, what is the value of $\chi^{(2)}$, where it comes from, how it depends on Fermi energy? No specific formula for $\chi^{(2)}$ is given in the paper. Figures 2(b,c) show that the gain very essentially depends on the Fermi energy, with a broad maximum at $E_F \sim 0.07$ eV and with strongly oscillating, resonant features around 0.4-0.5 eV. What are the physical reasons for these features? Which analytical formulas for $\chi^{(2)}$ demonstrate these feature? What is their physical meaning? The paper does not contain answers to these questions. Moreover, it seems that the authors do not know themselves which nonlinearity they are going to use in the proposed phenomenon. Just after Eq. (2) they write that " $\chi^{(2)}$ is the nonlinear susceptibility tensor describing all suitable second-order nonlinearities in the system". What are these "all suitable nonlinearities", are they due to graphene or to the waveguide material? Since the gain is plotted as a function of Fermi energy in graphene and, especially, the resonances are seen at $E_F \sim 0.4$ eV $= \hbar\omega/2$, this seems to suggest

that the nonlinearity comes from graphene. But in the Supplementary Note the authors write that the waveguide should at least partially be composed of a nonlinear material such as lithium niobate. What does it mean "partially composed" ? Do the authors have specific suggestions how the waveguide should be filled by the nonlinear lithium niobate? Are there specific calculations of the gain made with the LiNb-filled waveguide? Do these calculations show that the gain can be sufficiently high to make the whole story meaningful? Does all this mean that the plots shown in Figure 2 make no sense anymore?

9. In the next sentence from the Supplementary Note the authors write: "... we could not obtain an appreciable SPPS-gain based on this effect alone. The problem is that the polaritonic mode does not penetrate the waveguide enough to lead to a sufficient nonlinear mode overlap". This is just the central question of the whole story, why this important point is discussed somewhere in the Supplementary Note and not in the main text? Does this negative result mean that the SPPS cannot be actually realized? Does it make sense then to publish this paper?

10. In the next sentence from the Supplementary Note we read "We do, however, emphasize that this approach might prove very useful for the excitation of polaritons in materials other than graphene". But if, suddenly, the authors decide to replace graphene by other 2D materials, does this mean that all calculations and plots shown in Figure 2 make no sense anymore (they were done using specific properties, like $\sigma(\omega)$, of graphene)? For any other 2D material with a parabolic spectrum all specific results (plasma frequency and damping, nonlinear parameters, etc.) will be different.

11. Finally a couple of minor points:

(a) in the literature there exist several publications on the second order nonlinear response of graphene which are quite relevant for the discussed effect; not all of them are cited. Several publications of the Belyanin's group treated the plasmon-light nonlinear interaction in a much more comprehensive and quantitative way: PRL 112, 055501 (2014), Phys. Rev. B 93, 235422 (2016), Phys. Rev. B 94, 195442 (2016). The paper of the Sipe group, Scientific Reports 7, 43843 (2017), also treated the discussed nonlinearity and gave detailed derivations and quantitative results. In my opinion, the present paper does not add anything useful and new to these studies;

(b) the term "plasmon polariton" supposes that 2D plasmons strongly interact with light, i.e. the square-root dispersion $\omega \sim \sqrt{q}$ is close to the light line $\omega = cq$. In the present work the 2D plasmon frequency is always much smaller than the light frequency with the same q . Therefore the term "plasmon" (instead of plasmon polariton) and SPS (instead of SPPS) would be more relevant.

Summary: since the damping of 2D plasmons is many orders of magnitude larger than the damping of sound waves, the feasibility of SPPS had to be studied and clearly demonstrated in much more details than it is done in the paper. Personally I do not believe that the SPPS phenomenon will ever work in structures with graphene or any other 2D materials and I did not find in the paper convincing arguments that the proposed effect can be observed indeed. The paper is very badly written: the authors carefully hide from the reader details of the theory rather than explain what and how they did.

I do not recommend this manuscript for publication.

Comments of Reviewer 3 (italic font) and our responses

Reviewer:

The authors put forward an idea of “stimulated plasmon polariton scattering” (SPPS), to use in 2D materials like graphene, for studying fundamental physics and for some applications. The idea of SPPS is similar to the stimulated Brillouin scattering (SBS) phenomenon known in nonlinear optics. The authors claim that SPPS power gain in the proposed experiments and structures can be sufficiently large under achievable experimental conditions. The paper is theoretical.

Response:

This correctly characterises our work.

Reviewer:

In my opinion, it is quite possible that the idea is interesting and the proposed effects may work indeed. But I have not found in the paper clear evidences that the observation of the proposed effects is realistic.

Response:

We note that the reviewer does not dispute the novelty or potential impact of the idea, but questions that it can be achieved. In the following we will put forward the view that this impression is at least to some degree due to strong misunderstandings of both the proposed SPPS process, SBS and some aspects of nanophotonics in general. However, especially with the wide readership of Nature Communications in mind, we agree that every misunderstanding by a reviewer points at a potential weakness in the presentation such as implicit assumptions by the authors. Therefore, we have modified our manuscript in several places to clarify it based on the points raised by the reviewer in order to avoid misunderstandings by intended audience.

Reviewer:

There exists a very large difference between the SBS and the proposed SPPS phenomena. The propagation length of sound waves in liquids and solids can be as large as kilometers, while the propagation length of 2D plasmons in 2D materials is as small as just a few microns. Therefore the authors had to clearly show that the nonlinearity of the plasmon-light interaction is so large that it can overcome the strong damping of plasma waves in the system. This is the most important point and this had to be done with all details to convince the reader that the idea is realistic indeed. I have not found details

in the authors' arguments and their conclusions are not convincing at all.

Let me illustrate said above by some examples.

Response:

This really is the heart of the review and it is repeated in the summary at the end. Unfortunately, the reviewer is completely wrong in their understanding of the acoustic loss in SBS. Since it is their main point (as they say themselves), we think we should elaborate on this in more detail.

The reviewer names the propagation length as the relevant measure of loss, which we can agree to in the context of inelastic backscattering such as (backward) SBS and the proposed SPPS.¹ The propagation length l can be easily found from the quality factor Q and the acoustic/plasmonic wavenumber q :

$$l = Q/q.$$

The quality factor in turn can be found either as the ratio of wavenumber and decay constant (just the inverse of the equation above) or the ratio of acoustic frequency and damping rate. From the phase matching diagram Fig.1a, which is identical for both SBS and SPPS, it is clear that $q \approx 2k$, where k is the optical wavenumber and therefore very similar in both processes. This reduces the question of propagation lengths to comparing the quality factors Q of both processes. We find Q between 10 and 1000 in SPPS (see Fig.2b). As for the losses in SBS, we refer to a standard textbook [Boyd, "Nonlinear Optics, 3rd edition"], which on page 441 lists Stokes frequencies (denoted Ω_B) and damping rates (denoted Γ_B) in various bulk solids and liquids. We copied the table for convenience:

9.3. Stimulated Brillouin Scattering (Induced by Electrostriction) 441

TABLE 9.3.1 Properties of stimulated Brillouin scattering for a variety of materials^a

Substance	$\Omega_B/2\pi$ (MHz)	$\Gamma_B/2\pi$ (MHz)	g_0 (m/GW)	$g_B^d(\text{max})/\alpha$ (cm ² /MW)
CS ₂	5850	52.3	1.5	0.14
Acetone	4600	224	0.2	0.022
Toluene	5910	579	0.13	
CCl ₄	4390	520	0.06	0.013
Methanol	4250	250	0.13	0.013
Ethanol	4550	353	0.12	0.010
Benzene	6470	289	0.18	0.024
H ₂ O	5690	317	0.048	0.0008
Cyclohexane	5550	774	0.068	
CH ₄ (1400 atm)	150	10	1	
Optical glasses	15,000–26,000	10–106	0.04–0.25	
SiO ₂	25,800	78	0.045	

¹We note that there is some inconsistency in the opto-mechanical community and the term "SBS" is sometimes extended to another scattering process (usually known as guided acoustic waves Brillouin scattering, GAWBS) that involves a standing rather than propagating acoustic wave and occurs at lower frequencies with corresponding lower loss [Wolff et al. New J. Phys. **19**, 023021 (2017)]. However, there the loss does not come even close to values that the reviewer suggests, either.

By dividing Ω_B and Γ_B , one easily verifies that SBS has Q-factors in the same ballpark as SPPS. This also holds for material systems that have been the focus of recent SBS-research such as integrated soft glass waveguides ($\Omega/2\pi = 7.7$ GHz, $\Gamma/2\pi = 34$ MHz, [Pant et al., Optics Express 19, 8285 (2011)], corresponding to $Q = 226$) or silicon waveguides (e.g. [van Laer et al., Nature Photon. 9, 199 (2015)] reports $Q = 306$ in the main text).

We conclude that the strong loss of the plasmon is by no means more severe than that of GHz-sound in solids and liquids, which the reviewer seems to underestimate by 6-8 order of magnitude.

The reviewer criticises that we do not demonstrate that the opto-plasmonic nonlinearity is so much stronger than the opto-mechanical one that we can overcome the loss. The previous paragraphs should have shown that it is already sufficient if both nonlinearities are comparable in strength. The relevant parameter for such a comparison is the gain coefficient, which incidentally already includes the loss in the form of the decay constant κ_B (see Eq. 3). To reiterate: The gain coefficient G scales with the quality factor and therefore expresses precisely how strong the nonlinearity is compared to the loss of the non-optical wave (sound in SBS, plasmons in SPPS). The reviewer seems to have missed that the requested comparison was actually already in the manuscript, which we concede might be a at least partially a flaw in our presentation (see changes listed below).

We come to the question if the nonlinearity in SPPS can be strong enough to observe the effect and again we do this by comparison with the experimentally well-established effect SBS. In SBS the gain is stated in one of two ways: As a material parameter (units m/W) appropriate for quasi-bulk systems such as fibres or as a waveguide parameter (units $(Wm)^{-1}$) appropriate for complex nano-scale systems such as our example. The two gain figures are related by the effective mode area of the optical modes. Boyd lists the former quantity and based on an effective mode area of $0.1 \mu m^2$ (roughly what we have in our example), this translates to gains between $15 (Wm)^{-1}$ for CS_2 and $< 4.5 (Wm)^{-1}$ for water and silica. These values are considerably smaller than what we report. On page 4 of our original paper, we compare our gain figures to those found in technologically relevant systems such as optical fibres (which is in agreement with the rough estimate from Boyd's bulk parameter for silica) and integrated waveguides composed of soft glasses and silicon:

We find Stokes shifts throughout the entire THz-range, and for low Fermi-levels we predict SPPS-gains in excess of $10^4 (Wm)^{-1}$ over a fairly large parameter range with peaks approaching $10^6 (Wm)^{-1}$. This has to be compared to $1-10 (Wm)^{-1}$ for SBS in optical fibres [29,30], $100-1000 (Wm)^{-1}$ in chalcogenide rib waveguides [16,31] and $1000-10000 (Wm)^{-1}$ in well-engineered silicon waveguides [32-34].

We believe that this response has demonstrated that the reviewer's main complaint is not warranted. Plasmonic loss is not more damaging to SPPS than the acoustic loss is to SBS and the opto-plasmonic nonlinearity that we calculate here is not weaker than the opto-mechanical nonlinearity found in SBS.

Since the plasmonic loss is such an important point to the reviewer and since we now fear that readers from outside the area of SBS might also have an inaccurate idea of the the acoustic loss in SBS, we have modified the main text to address the problem of plasmonic loss as early after the introduction as possible in order to prevent this confusion to arise in the first place:

In analogy to SBS, overall conservation of energy and momentum require that both the polariton field as the optical Stokes wave grow approximately exponentially along the waveguide. This process exists as soon as there is an efficient nonlinear coupling mechanism between the optical pump and the polariton. Naturally, this nonlinear coupling must be strong enough that the process is not immediately quenched by the linear losses experienced by the plasmon polariton. At least for THz graphene plasmon polaritons this is actually not as bad as it might perhaps sound. Their propagation loss is in fact very comparable to that of GHz-range acoustic phonons found in SBS as illustrated by the fact that both THz graphene plasmons as well as GHz sound waves in technologically relevant materials [14,16,31--33] have quality factors between 10 and several 1000. Finally, neither SBS nor SPPS require long-range propagation of their respective non-optical excitations.

We have also added a new section (section IV) to the supplementary material paraphrasing our response to this question (not reprinted here to save space and because it is very similar to the previous few paragraphs).

Furthermore, we modified the manuscript to explicitly state that the gain expresses the relative strength of nonlinear coupling and plasmonic loss:

In analogy to the theory of SBS [26] we can easily derive from Eqs. (1a--1c) the stationary power gain

$$G = \frac{2\omega\Omega|Q|^2}{\Gamma_a^2\Gamma_b\kappa_b} \quad (1)$$

which is the most interesting quantity in experiments. Besides inconsequential normalization constants and the frequencies ω , Ω , its main ingredients are the nonlinear coupling integral Q and the polaritonic loss parameter κ_b . Therefore the reader concerned with loss in graphene may interpret G as a measure for how strong the nonlinear coupling is compared to the linear polaritonic loss. This should be kept in mind when comparing the gain in successful SBS experiments to the numerical values that

we calculate further below. The derivation of these equations and those underlying our numerical example is beyond the scope of the main text and provided as a supplementary note.

Finally, we realised while writing this response that readers who turn towards the literature might be confused by the two different ways in which SBS-gain is expressed and added a footnote:

In order to avoid confusion we note that SBS-gains for bulk materials are often specified in alternate units of m/W . Gains specified in $(\text{Wm})^{-1}$ are more suited to nano-scale waveguides and differ from the former by the effective waveguide mode area.

Reviewer:

1. In Figure 1 the authors show schematic of the SPPS experiment: a rectangular dielectric optical waveguide is covered by a graphene sheet and the nonlinear plasmon-light interaction should lead to the desired effect. The general idea is clear but there are some questions to its practical realization. In particular, we read "A weak THz signal is injected through the bias contact (blue) and amplified along the waveguide". How the THz wave propagating in free space (wavelength hundred of μm) will be injected into the waveguide with dimensions 200 nm? I believe that a lot of energy will be lost on the interface between the tens-of- μm THz antenna and sub- μm optical waveguides. This issue is not discussed in the paper.

Response:

Fig.1 is a conceptual cartoon. All three panels are meant to give a high-level picture of what is going on: Fig.1a illustrates the energies and momenta of the waves, Figs.2b/c illustrate which waves propagate and/or grow in which direction, which ones are injected and what comes out. The box-like structure is meant to represent a photonic waveguide on a conceptual level, it is not meant to indicate which geometry might be beneficial; this is left to our example waveguide shown in Fig.2. In the caption of Fig.1, we write

``Schematic of an SPPS-experiment in Stokes-seed configuration,``

``Schematic of an SPPS-experiment in polariton-seed configuration,``

for Fig.1b and Fig.1c, respectively. We believed that this already captured our intent. Nonetheless, we made it a little more specific and amended the caption of Fig.1 (see below).

In Fig.1 and in the main text, we propose two basic modes of operation. The first mode is analogous to what is called a "Brillouin-amplifier" by some in the SBS community, where a weak optical

Stokes seed is amplified via the inelastic scattering process. This mode of operation does not require the injection or direct detection of THz waves. Instead, it requires only two laser around an arbitrary wavelength (e.g. a telecom band) detuned by the Stokes shift and optical components that operate at the laser wavelength. Because of this and because the “amplifier”-configuration is usually also the first experiment to show SBS in nanoscale waveguides, this would be the natural mode for a demonstration experiment. The reviewer’s concerns do not apply to this configuration.

The second mode is more similar but not quite identical to the “Brillouin-generator” scheme from SBS, where preexisting sound/plasmons are amplified along the waveguide. The reviewer is correct that technical challenges lie ahead in implementing this, but this is not really surprising, as it is also harder to achieve in SBS. Nonetheless, we regard this configuration as noteworthy, because of its potential benefits.

In response to this, we have further clarified in our manuscript which configuration would be the first choice for initial experiments and which one includes greater technical challenges:

(b) **Schematic of an SPPS-experiment in Stokes-seed configuration.** Conceptual schematic of an SPPS-experiment in Stokes-seed configuration. A weak Stokes seed (red arrows) is injected at the rear end and amplified as it propagates towards the front end of a graphene-covered waveguide. This action is also illustrated by qualitative spectra where the heights of the colored peaks indicate to the relative amplitudes of the matching signals at the front and back of the waveguide. **The bias voltage contact and the waveguide geometry shown here were intentionally kept oversimplified at this stage and in reality would require careful engineering.** This configuration is best suited for the excitation of polaritons and for characterizing their dispersion relation, which can be tuned via the bias voltage contact. **It is also the natural candidate for a first demonstration of SPPS as all input and output signals are in the optical domain and no injection or detection of THz waves is required.** (c) **Schematic of an SPPS-experiment in polariton-seed configuration.** Conceptual schematic of an SPPS-experiment in polariton-seed configuration. A weak THz signal is injected through the bias contact (blue) and amplified along the waveguide. Simultaneously, an optical Stokes is generated (illustrated by qualitative spectra similar to panel b). **This configuration is best suited to amplify or optically detect weak THz signals. This configuration is compelling because of the prospect to amplify or optically detect weak THz signals.**

Reviewer:

2. In Figure 2(a) the authors suggest two possible realizations of the device. Now we see two dielectric waveguides instead of one from Figure 1. Why two waveguides are better than one? In my opinion, if the graphene sheet lies on the waveguide the light-plasmon inter-

action (the integral in Eq. (2)) will be larger than when a piece of graphene hangs over the gap of 150 nm width (sheet position 2); this hanging piece of graphene will contribute to the plasmon damping and will work against the desired effect. The sheet position 1 causes even more questions. The optical field is mainly concentrated inside the waveguides and substantially decreases in the gap between the waveguides which is seen from the authors' CONSOL calculations. Placing graphene between the two waveguides will reduce the interaction again as compared to the structure in Figure 1. Why the authors chose the two-waveguide structure is not discussed in the paper. If my intuitive arguments are wrong and the authors did perform calculations of the field distributions in and around the waveguides and found the best positions for the graphene sheets indeed, why these calculations are not shown and not discussed in detail in the paper?

Response:

The reviewer fundamentally misunderstands the proposed waveguide geometry. We do not propose two waveguides, but one slot waveguide composed of two closely spaced silicon beams. This is by no means a novelty, it is in fact a well-known waveguide layout in nonlinear photonics due to its strong field concentration inside the slot. This comment made us aware that we had implicitly assumed that every reader would be familiar with this principle, which is apparently not true.

Much of the reviewer's discussion is based on their misunderstanding of the slot waveguide. The "hanging" part is in fact where most of the nonlinear coupling occurs due to the concentration of the photonic mode and moving the sheet further into the gap center improves this nonlinear coupling (while also changing other aspects of the system such as the plasmonic dispersion relation and the linear waveguide loss). The reviewer left us confused when they first say that our COMSOL calculations (Fig.2a) show weaker field in the gap compared to the silicon bulk (which they most certainly do not), but then criticise that we did not perform any field calculations in the first place. Nonetheless, we realised that the inclusion of a qualitative colorbar in Fig.2a might have helped the review in recognizing the field enhancement inside the slot, so we added it.

We believe that slot-waveguides are so well-established at this point that a detailed discussion of their field patterns would be inappropriate for Nature Communications. Nonetheless, we have added a brief statement explaining why we chose this particular geometry. In response we added a brief explanation and reference to the literature (we also fixed a typo):

As a waveguide, we select a slot waveguide composed of two silicon beams with square rectangular square cross section (dimension $220 \times 275 \text{ nm}^2$) separated by a 150 nm air gap and operated at the standard telecom

wavelength 1550 nm corresponding to a photon energy of 0.8 eV. We computed the electric field distribution of the fundamental mode (effective index 1.37) using the commercial finite element software COMSOL (see Fig. 1a). The slot waveguide was selected, because this configuration is known to support waveguide modes that are mostly confined in the gap between the silicon beams and are therefore an excellent starting point for nonlinear photonics in the silicon nanophotonics platform [?].

Reviewer:

3. It seems that the difference between positions 1 and 2 is in the dielectric constant of the environment, $\epsilon = 1$ in the position 1 and $\epsilon = (1 + \epsilon_{\text{substrate}})/2$ in the position 2: due to this difference the plasmon frequency and the quality factor are higher in the position 1 than in the position 2 (unfortunately I have to guess here since the authors did not explain this point). However in typical experiments graphene lies on a substrate. If the authors imply just freely hanging graphene in the position 1 I am wondering how this can be practically realized in a real device? If in the position 1 graphene lies on a substrate than the advantage of this geometry ($\epsilon = 1$) disappears.

Response:

The reviewer correctly states that the two positions differ in the dielectric environment (known as dielectric loading), which explains e.g. the different Stokes shifts found for them. However, the (in our opinion) more obvious and relevant difference is the higher optical field intensity in the center of the slot, which the reviewer seems to have missed, probably because of their misunderstanding of the slot waveguide geometry. On page 5 of the original manuscript, we briefly discuss this topic for the different findings for positions:

[. ..]

Furthermore, we point out that any form of field enhancement will improve the relative gain figure F . While it is true that placing the polaritonic sheet in higher field enhancements inevitably increases the linear loss due to inter-band transitions, the SPPS process (like SBS and Raman scattering) is effectively a third-order nonlinearity and thus is bound to over-compensate the increase in loss.

Secondly, we find that dielectric loading of the polaritonic mode is quite beneficial. This is the reason why a graphene sheet positioned above a slot waveguide provides gains that are comparable to that of a narrow ribbon squeezed inside the gap, even though it is subject to considerably lower intensity enhancement and overall poorer mode overlap. It is not clear how screening of the polaritonic mode (e.g. due to the use of gold-plasmonic waveguides) affects the overall gain.

While this discussion was kept concise, we believe it does contain the main message. Furthermore, we believe that these two paragraphs from the original manuscript also address much of the reviewer's point 2.

Reviewer:

4. In Figure 2(a) the waveguide material is assumed to be Si. But Si is a semiconductor often doped by some amount of impurities. This will lead to an additional damping of 2D plasmons and maybe to a damping of the optical modes. The propagation of optical modes in the Si waveguides is also not explained. The only information that the effective index equals 1.37 is insufficient; moreover this number seems to be too low for Si waveguide (the dielectric constant of Si is 11.7). Usually experimentalists use dielectric materials for the waveguides like silicon oxide or silicon nitride. Why semiconducting silicon is supposed to be used for SPPS in this work?

Response:

We believe that silicon is well established as a waveguide material in the near-IR (see e.g. [Saleh/Teich "Fundamentals of Photonics"] or other textbooks) and it is in fact so mature that institutions like IMEC in Ghent exist. Linear absorption of silicon is not of major concern in the near-IR; surface roughness is often more critical, at least in slot waveguides.

We are quite unsure what information about the propagation of the optical modes the reviewer expects. The only additional information that might be useful (but somewhat obvious from the field plot and easily verified via the effective mode index) is that we consider the horizontally polarized mode.

We cannot follow the reviewer's comments on the link between the dielectric constant of silicon (which they correctly identify around 11.7) and the effective mode index. The latter is more or less an average of all permittivities seen by the optical mode. For a mode that mostly lives in the surrounding air such as ours, it will naturally be close to 1.

We are confused by the remarks that waveguides are usually made of silicon nitride and especially in the case of silica.

Reviewer:

0. The authors assume to change the Fermi level from <0.01 eV up to <2 eV (Figure 2(b)). But, firstly, it is unrealistic to push the Fermi level in graphene above <0.4 eV, therefore the very high quality factors and very broad frequency ranges "achieved" at Fermi energies up to <1 eV should be reduced by several times or even by an order of magnitude in realistic structures. Secondly, changing the

Fermi level assumes that graphene lies on a SiO₂/doped Si substrate and E_F is varied by the gate voltage applied between graphene and Si. But then, if the hanging graphene is assumed to be in the sheet position 1, all left plots drawn in Figures 2(b,c) are useless.

Response:

The reviewer is correct that electrostatic doping beyond 0.4-0.5eV is increasingly difficult. However, recent discussions with peers have indicated that at least values up to 0.5eV are not unrealistic. We decided to show data for higher energies anyway, because readers might find a way to reach them e.g. via chemical doping.

Furthermore, the reviewer is correct that gating of this structure cannot be simply done with a simple back-electrode, as we have learnt in discussions with peers. It is, however not impossible; a metallic gate 1-2 microns above the structure should do the job at the expense of added complication. An alternative possibility would be top-gating using ion gels (see e.g. [Ju et al., “Graphene plasmonics for tunable terahertz metamaterials,” Nature Nanotechnology 6, 634 (2011)]). A third possibility would be gating via direct metallic contacts to the graphene sheet (similar to [Ding et al., Nano Lett. 15, 4393 (2015)]) which we alluded to in the cartoonish way we apply “bias” in Fig.1b and Fig.1c. Coincidentally, this last reference actually demonstrates experimentally the drop in the linear loss of a silicon waveguide covered with a graphene sheet as the Fermi level is moved towards the inter-band threshold at 0.4 eV. We added this paper as a reference to our manuscript. Finally, while electrostatic doping would clearly be the preferred doping mechanism, it is of course not the only option for a demonstration experiment. Chemical doping would be harder to control, but otherwise just as effective and avoid some of the technological problems. We have modified the manuscript to reflect this:

We will now apply this theoretical framework to an illustrative waveguide geometry that has not been optimized for the strongest possible gain, but rather selected for its simplicity. The main purpose of this example is to demonstrate that the concept is feasible, i.e. that sufficient gain can be achieved in a realistic setting. We focus entirely on the interplay of the opto-plasmonic nonlinearity and linear losses and leave important technological aspects of an actual experiment out. This includes especially the exact method of doping, which can be achieved e.g. electrostatically via a metal contact few microns above the structure, with direct electrical contacts [?] or by chemical doping.

Ultimately, we are willing to argue over which level of detail is required in a fundamental feasibility study. The main point of Fig.2 is to demonstrate that the nonlinearity can be made sufficiently strong while leaving other details such as gating and in- and out-coupling of light open. In our opinion, this should be acceptable in this type of publication.

The reviewer is oddly focused on back-gating via the silicon beams. This would indeed require heavily doped silicon and would be extremely lossy. However, we never suggested this anywhere in the paper.

Finally, the reviewer implies that our most interesting results appear for doping levels beyond 0.4-0.5eV, which is not quite true. The highest gain values are found for *low* Fermi energies (see Fig.2b). The highest relative gain (i.e. best parameter range for experimental observations) is found just above 0.4eV, which according to the reviewer is still feasible.

Reviewer:

6. If graphene lies on a SiO₂/doped Si substrate (in order to control E_F) what is the 2D plasmon spectrum in this paper? As far as I understand (I have to guess again since the authors do not discuss this point) the authors assume the square-root dispersion of 2D plasmons, $\omega < q$; this can also be seen from the sketch in Figure 1(a). But in the graphene-dielectric-metal (or heavily doped semiconductor) system the spectrum of 2D plasmons is linear, $\omega^2 < q^2 d$ at $qd < 1$, where d is the dielectric thickness. If to take this fact into account the plasmon frequency and the quality factor will be further reduced. This important practical issue is not discussed in the paper.

Response:

The reviewer criticises that we use the square-root dispersion relation. They very correctly point out that this assumption would be unacceptable if the graphene sheet were placed on a metal or a heavily doped semiconductor. We never suggest anything of the kind. We propose to place it on or between undoped silicon. The fact that we take the dielectric loading due to silicon into account (see the discussion regarding page 5 mentioned in point 3):

Secondly, we find that dielectric loading of the polaritonic mode is quite beneficial. This is the reason why a graphene sheet positioned above a slot waveguide provides gains that are comparable to that of a narrow ribbon squeezed inside the gap, even though it is subject to considerably lower intensity enhancement and overall poorer mode overlap.

should suggest to the reader that we include the effect of nearby dielectrics. We mention metals only in passing as an outlook for further work:

Curiously, we find the opposite effect in SPPs: the polaritonic quality factor *increases* as the wave number and frequency are increased. We believe this to be a direct result of decreasing polaritonic mode confinement. As a result, high-mode-index optical waveguide are more viable and

especially the use of plasmonic waveguides based on noble metals appears quite interesting.

We don't mention heavily doped semiconductors anywhere.

Reviewer:

7. The authors give minimal information about how the frequency of 2D plasmons (Stokes shift) and their propagation length (quality factor) are plotted in Figure 2(b). No one formula is given here. They write "The polaritonic damping parameter γ_c depends on E_F . We calculate it from a Drude relaxation rate that was experimentally determined for this type of "vdW-sandwich" [24] and plot it implicitly in Fig. 2b as the polaritonic quality factor $= \Omega/(2v_b\gamma_c)$ ". Why the authors did not give here just two numbers, the relaxation rates for 60 K and 300 K, which they extracted from [24]? Why should I or other readers read the long paper [24] only in order to find numbers which the authors used in their paper? I have found the number of 1.6 ps for graphene at 60 K and could get a figure similar (but not identical, again due to the lack of information) to that shown in Figure 2(b) but I still have no idea how the authors plotted the curves for 300 K. Why should the readers guess what the authors' thoughts were and not just read what they really did?

Response:

We are more than happy to include the damping parameters that we used in our calculation. We also added a statement explaining why we chose to plot the loss the way we did:

As a polaritonic system, we select a graphene monolayer that has been sandwiched between thin layers of hexagonal boron nitride to isolate it from adverse substrate effects. The polaritonic damping parameter γ_c depends on E_F . We calculate it from a Drude relaxation rate that was experimentally We calculate it from the Drude relaxation rates (25meV at 300 K and 0.25meV at 60 K) that were experimentally determined for this type of "vdW-sandwich" [?] and plot it implicitly in Fig. 2b as the polaritonic quality factor $\Omega/(2v_b\gamma_c)$. As mentioned before, this quality factor is the appropriate measure when comparing to loss found in SBS and other related scattering processes.

Reviewer:

8. It is unclear at all how the authors calculated the SPPS gain (2) and plotted the corresponding Figures 2(b,c). The value Q in (2) depends on the overlap integral and the nonlinear susceptibility $\chi^{(2)}$. In the SBS effect the interaction takes place in the whole (3D) volume of the waveguide. In the considered here SPPS effect the integration goes over a much smaller volume (over just a 2D plane if $\chi^{(2)}$ is due to the nonlinearity of graphene).

Response:

We calculated the gain by numerically evaluating the integrals stated in Eqs.(5,6,20,26,35) of the supplementary material.

We are not quite sure about the statement regarding 3d and 2d integrals. They appear to believe that the nonlinearity must be small because the graphene sheet is very thin, but this kind of reasoning seems flawed to us. A single atom on resonance for example has an extinction cross section on the order of the wavelength of light, so why wouldn't the electrons in a 2d material? In fact, one major appeal of graphene and other 2d materials is the fact that they do very strongly interact with light despite being atomically thin. This includes the absorptivity of graphene or the recent research on strong coupling of 2d material excitons and photonic resonators. As mentioned in our manuscript:

If the Fermi level is below half the photon energy (here: 0.4 eV), graphene is extremely lossy due to inter-band transitions, leading to high linear waveguide loss (in our example reaching power loss values up to 10000 dB/cm).

If undoped graphene can introduce such high optical loss (which is why it has been proposed as a material for modulators [Ding et al., Nano Lett. 15, 4393 (2015)]), why should it not also introduce a strong nonlinearity in a low-loss regime? To clarify: This is not speculation on our part regarding the nonlinear effect from graphene. We merely wish to highlight the conflict between the reviewer's positions "graphene introduces prohibitive amounts of linear loss" and "graphene is too thin to matter for the nonlinear response."

Reviewer:

Here the authors had to show that, in spite of the thin interaction area the resulting gain is sufficient to overcome losses. But no results of such analysis are shown.

Response:

We don't know what kind of analysis the reviewer had in mind. To us the problem is quite straight-forward: We numerically calculated the integrals and other quantities that make up the gain coefficient. The numerical values of these intermediate results are quite meaningless, because they depend on details such as the mode normalization, so we don't present them. The meaningful results (gain, Q-factor, Stokes shift) have been plotted in Fig.2b. As explained before the gain coefficient already expresses the relative strength between nonlinear coupling and linear plasmonic losses. We then proceed to relate that to the linear optical loss of the waveguide modes via the relative SPPS gain F (shown in Fig.2c). Since we are talking about a nonlinear gain process competing with linear loss, the nonlinear gain will always

be stronger than the linear at sufficiently high pump power. The question is just how high that pump power has to be. This is answered by the relative gain F , as we write in the manuscript:

Since the linear optical loss limits the useful waveguide length, a natural figure of merit is the ratio between gain and loss:

$$F = G/\kappa_a, \quad (2)$$

which takes units of W^{-1} and whose inverse $1/F$ is the pump power level necessary to achieve net gain (the effect can remain detectable at considerably lower pump levels e.g. by pump modulation and lock-in amplification of the output Stokes). We plot this in Fig. ??c. For most Fermi levels, we find values around $0.1 \dots 1 W^{-1}$, which is enhanced up to $100 W^{-1}$ due to a ponderomotive resonance at the inter-band threshold. This means that peak pump powers from $10 W$ down to $10 mW$ would provide net-gain (the latter at liquid nitrogen temperatures and for waveguide lengths in the low millimeter range). This is very feasible with state-of-the-art silicon photonics techniques and sub-ns pulsed light sources would prevent damage due to excessive dissipation in the graphene and free carrier absorption in silicon.

We are not sure what fundamentally different we can do but to calculate the gain G and compare it to the gain of a process that is known to work (SBS) and then to check whether the required pump powers are realistic. While the $10 W$ peak power is a bit of a stretch in such a small silicon structure, $10 mW$ is no problem even in cw. Pulsed experiments should be feasible between these two bounds.

Reviewer:

Then, what is the value of $x^{(2)}$, where it comes from, how it depends on Fermi energy? No specific formula for $x^{(2)}$ is given in the paper.

Response:

The reviewer must have missed our statement

In our example system, the second order nonlinearity stems entirely from the intrinsic ponderomotive force of the electron system in graphene.

as well as the section “Ponderomotive nonlinearity in graphene” in the supplementary material, which contains a brief derivation of the nonlinearity. A full derivation might constitute a case of self-plagiarism, as we published it earlier this year in the New Journal of Physics (Ref.[28]). We note that Ref.[28] was available on the arXiv throughout the review process, so we did not hide anything in this regard.

After some thought (and being reminded of some discussions with peers and also the review process for our Ref.[28]) we notice that the

reviewer consistently writes about the $\chi^{(2)}$ -nonlinearity of graphene. Initially, we read this as a slightly inaccurate synonym for second-order nonlinearity in general (as we do ourselves sometimes) and attached no further significance to it. However, upon second thought, it might actually point at a second critical misunderstanding.

We would like to stress that we are *not* using the Pockels nonlinearity $\chi^{(2)}$ of graphene, but the ponderomotive nonlinearity. While both are second order nonlinearities, they have very different symmetries. The nonlinear Pockels susceptibility $\chi^{(2)}$ describes the interaction between in general three electric field vectors (hence being described by a third-rank tensor), whereas the ponderomotive nonlinearity relates the optical intensity with the local charge density and is therefore in general described by a second rank tensor. The hexagonal symmetry of graphene reduces this to a scalar when only the in-plane electric field is taken into account. Since this might turn out to be a common, somewhat subtle but nonetheless consequential misunderstanding, we have eliminated our inaccurate use of the symbol $\chi^{(2)}$ in the supplementary material (not quoted here to save space and because it just involves replacing mathematical symbols) and in the main text:

The interaction is mediated by the nonlinear overlap between the optical pump and Stokes modes and the permittivity change caused by the polaritonic mode:

$$\begin{aligned}
 Q &= \int \mathbf{e}^{(p)} \cdot \Delta \mathbf{E}(\mathbf{e}^{(pol)}) \mathbf{E}^{(s)} \\
 &= \int d^2A E_0^p E^s \int_{ijk} [e^{(p)}]_i^* e^{(s)}_j e^{(pol)}_k \chi^{(2)}_{ijk}, \\
 &= \int d^2A E_0^p E^s \int_{ijk} [e^{(p)}]_i^* e^{(s)}_j e^{(pol)}_k \Delta E_j(\mathbf{e}^{(pol)}),
 \end{aligned} \tag{3}$$

where the integral is carried out over the cross sectional plane of the waveguide, $\mathbf{e}^{(pol)}$ is the electric field distribution of the polariton mode and $\chi^{(2)}$ is the nonlinear susceptibility tensor describing all suitable second-order nonlinearities in the system. $\Delta E_j(\mathbf{e}^{(pol)})$ includes all suitable second-order nonlinearities in the system. This may include conventional Pockels nonlinearities $\chi^{(2)}$ in the waveguide or the polaritonic material or effects such as the ponderomotive nonlinearity, which we base our example on.

Furthermore, we have reiterated this difference where we introduce the example system in the main manuscript:

In our example system, the second order nonlinearity stems entirely from the intrinsic ponderomotive force of the electron system in graphene. It is a direct consequence of the dependence of the optical properties on the Fermi level and has been studied based on a pure intra-band model for the optical conductivity [30]. In addition, we also consider

the inter-band contributions and finite-temperature corrections to

the ponderomotive force that we will present in detail in an upcoming paper [31]. We have published elsewhere [31]. We should stress that while the ponderomotive interaction constitutes a second-order nonlinearity, it must not be confused with the second-order nonlinear electric susceptibility $\chi^{(2)}$. The latter is a third-rank tensor relating three electric field vectors, and the former is sensitivity of the conductance tensor to variations in the scalar carrier density and therefore in general described by only a second-rank tensor.

Reviewer:

Figures 2(b,c) show that the gain very essentially depends on the Fermi energy, with a broad maximum at $E_F < 0.07$ eV and with strongly oscillating, resonant features around 0.4-0.5 eV. What are the physical reasons for these features? Which analytical formulas for $\chi^{(2)}$ demonstrate these features? What is their physical meaning? The paper does not contain answers to these questions.

Response:

Yes, the ponderomotive nonlinearity depends strongly on both E_F and ω with a pronounced resonance near the interband threshold. Yes, this does have physical reasons and is quite interesting indeed. It is however irrelevant for the discussion at hand and can be found in the separately published paper Ref.[28].

Reviewer:

Moreover, it seems that the authors do not know themselves which nonlinearity they are going to use in the proposed phenomenon. Just after Eq. (2) they write that " $\chi^{(2)}$ is the nonlinear susceptibility tensor describing all suitable second-order nonlinearities in the system". What are these "all suitable nonlinearities", are they due to graphene or to the waveguide material?

Response:

The answer is quite simple: Every second-order nonlinearity that leads to a sufficiently large integral is suitable, which might require careful alignment between waveguide mode polarizations and crystal axes or other symmetry considerations similar to those derived in [Wolff et al, Optics Express 22, 32489 (2014)] for the opto-mechanical nonlinearity in SBS. The fact is that we did not want to get bogged down with this and leave a group theoretical analysis of the nonlinear interaction as future work for a more technical paper. Instead, we decided to base our example entirely on the ponderomotive non-linearity of graphene, which is suitable in the aforementioned sense.

Reviewer:

Since the gain is plotted as a function of Fermi energy in graphene and, especially, the resonances are seen at $E_F < 0.4\text{eV} = \hbar\omega/2$, this seems to suggest that the nonlinearity comes from graphene. But in the Supplementary Note the authors write that the waveguide should at least partially be composed of a nonlinear material such as lithium niobate. What does it mean "partially composed" ? Do the authors have specific suggestions how the waveguide should be filled by the nonlinear lithium niobate? Are there specific calculations of the gain made with the LiNb-filled waveguide? Do these calculations show that the gain can be sufficiently high to make the whole story meaningful? Does all this mean that the plots shown in Figure 2 make no sense anymore?

Response:

We cite from the original manuscript:

In SPSS, the excitation is a localized polariton and the nonlinear interaction is either due to an intrinsic nonlinearity of the polariton system or to a Pockels nonlinearity in the waveguide.

[...]

This process exists as soon as there is an efficient nonlinear coupling mechanism between the optical pump and the polariton.

[...]

We will now apply this theoretical framework to an illustrative waveguide geometry that has not been optimized for the strongest possible gain, but rather selected for its simplicity.

[...]

In our example system, the second order nonlinearity stems entirely from the intrinsic ponderomotive force of the electron system in graphene. It is a direct consequence of the dependence of the optical properties on the Fermi level and has been studied based on a pure intra-band model for the optical conductivity [27]. In addition, we also consider the inter-band contributions and finite-temperature corrections to the ponderomotive force that we will present in detail in an upcoming paper [28].

We believe that there is no need for speculation where the nonlinearity comes from in our example (note: the "upcoming paper" has undergone peer review in the meantime and was published in the New J. Phys).

Regarding the use of lithium niobate, we believe it is most useful to quote the entire the paragraph in question, especially because in the following the reviewer picks individual sentences from it:

The most natural source for a second order nonlinearity in the system is of course the Pockels effect. This can be introduced, e.g. by composing the waveguide at least partially of a nonlinear material

such as lithium niobate. We cannot say much more about such a setup except that we could not obtain an appreciable SPPS-gain based on this effect alone. The problem is that the polaritonic mode does not penetrate the waveguide enough to lead to a sufficient nonlinear mode overlap. We do, however, emphasize that this approach might prove very useful for the excitation of polaritons in materials other than graphene.

At least in the case of graphene the dominant second-order nonlinearity for the SPPS-process turns out to be the ponderomotive interaction, i.e. the local shift of the Fermi energy inside the graphene sheet as a result of the beat between the optical fields and -- conversely -- the emergence of a dynamic grating in the graphene conductance from a plasmon polariton.

This segways into the section devoted to the ponderomotive nonlinearity in graphene. The paragraph in question is not aimed at the specific example discussed in the main text, but on a more conceptual level it points out what sources of nonlinearity might work. We also point towards the challenges that we have encountered in finding a suitable geometry, which again might come handy to the reader.

Reviewer:

9. In the next sentence from the Supplementary Note the authors write: ". . . we could not obtain an appreciable SPPS-gain based on this effect alone. The problem is that the polaritonic mode does not penetrate the waveguide enough to lead to a sufficient nonlinear mode overlap". This is just the central question of the whole story, why this important point is discussed somewhere in the Supplementary Note and not in the main text? Does this negative result mean that the SPPS cannot be actually realized? Does it make sense then to publish this paper?

Response:

No, this is not the central point of the story and it is not a particularly important point. It appears only as a minor comment in the supplementary, because as far as we are concerned it is nothing but a minor comment. To paraphrase, we say: "a waveguide composed of lithium niobate is not sufficiently nonlinear on its own to show SPPS with graphene and we know why.". It is not essential for the message of the paper and it most certainly does not mean that SPPS cannot work in principle. Our results in the main text show that the intrinsic nonlinearity of graphene should be plenty strong enough.

Reviewer:

10. In the next sentence from the Supplementary Note we read "We do, however, emphasize that this approach might prove very useful"

for the excitation of polaritons in materials other than graphene". But if, suddenly, the authors decide to replace graphene by other 2D materials, does this mean that all calculations and plots shown in Figure 2 make no sense anymore (they were done using specific properties, like $\alpha(\omega)$, of graphene)? For any other 2D material with a parabolic spectrum all specific results (plasma frequency and damping, nonlinear parameters, etc.) will be different.

Response:

The calculation in the main text is an example that is meant to show that the SPPS process (the SBS-like scattering between light and propagating polaritons) can be observed in at least one system (graphene on/in silicon slot waveguide). It does not become incorrect just because there might also be other possible realisations of the same basic idea. In this sentence, we simply point out that while SPPS with graphene requires the intrinsic nonlinearity of graphene, this might not be necessary for SPPS with plasmon-like excitations in other 2d materials and we invite fellow researchers to look for such alternatives. Of course they would find different gains, Q-factors and Stokes-shifts, just like when studying SBS in different materials or even just different waveguide geometries. We are confused why this is supposed to be a problem with our paper.

Reviewer:

11. Finally a couple of minor points: (a) in the literature there exist several publications on the second order nonlinear response of graphene which are quite relevant for the discussed effect; not all of them are cited. Several publications of the Belyanin's group treated the plasmon-light nonlinear interaction in a much more comprehensive and quantitative way: PRL 112, 055501 (2014), Phys. Rev. B 93, 235422 (2016), Phys. Rev. B 94, 195442 (2016). The paper of the Sipe group, Scientific Reports 7, 43843 (2017), also treated the discussed nonlinearity and gave detailed derivations and quantitative results. In my opinion, the present paper does not add anything useful and new to these studies;

Response:

First we note that the mentioned PRL was already cited in our manuscript. The reviewer does not explicitly claim we didn't, but suggests it. Next, we point out that the references proposed by the reviewer did *not* treat the nonlinearity studied in our manuscript (which is why the Scientific Reports was initially not cited). The listed papers differ from our work in two main ways: Firstly they study the Pockels nonlinearity rather than the ponderomotive nonlinearity. This is an important detail. Secondly and more fundamentally they do not consider the effect of self-stimulated inelastic scattering. The main point of SPPS (and SBS) is the fact that a sin-

gle Stokes photon generated at one end of the waveguide is subject to coherent resonant gain along the entire waveguide, which dramatically boosts the nonlinearity. This has three specific requirements: That the Stokes-shifted optical signal is kept inside a waveguide so it can further coherently interact with the pump, that it propagates towards the pump and that a strongly confined “idler” is fully phase-matched. If they are met then the nonlinearity is strongly enhanced (for example to the point that SBS limits the optical power in fibre-communication because even a single thermal phonon can cause an “optical avalanche” that back-scatters the entire optical signal). In this sense, our work treats the problem on a different level of detail: Not the details about the origin of the nonlinearity in graphene, but rather the consequences that such a nonlinearity will have on a long waveguide and how this would be useful for the design of devices. While most of the aforementioned prerequisites can be found in the literature when looking for them, we note that the important consequence of self-amplification had not drawn, yet. This should be “new and useful”, especially because the principle is not restricted to graphene on silicon, but applies to a much broader class of material systems.

We are happy to follow the reviewer’s advice and elaborate more on how our work relates this field of research. We added a paragraph to the introduction and added the suggested paper by Cheng, Vermeulen and Sipe:

The proposed mechanism bears similarities with the excitation of graphene plasmons [10] through difference-frequency generation (DFG) via the intrinsic Pockels nonlinearity of graphene [26], but our aim is different. Our focus is not the intrinsic graphene nonlinearity that unlocks DFG, but rather the consequences of this nonlinearity on the interplay between optical and polaritonic fields that are confined in a common long waveguide. We show that this will cause exponential gain for the THz polaritonic wave on a length scale well beyond its linear propagation length provided the nonlinear interaction is sufficiently strong. We find that among the nonlinear processes that can be found or introduced in a system composed of a dielectric waveguide combined with a graphene sheet, the ponderomotive nonlinearity [27] is sufficiently strong, but we stress that it is not necessarily the only possibility.

Reviewer:

(b) the term “plasmon polariton” supposes that 2D plasmons strongly interact with light, i.e. the square-root dispersion $\omega < (q)$ is close to the light line $\omega = cq$. In the present work the 2D plasmon frequency is always much smaller than the light frequency with the same q . Therefore the term “plasmon” (instead of plasmon polariton) and SPS (instead of SPPS) would be more relevant.

Response:

We kind of agree with the reviewer. Initially, we actually did use the abbreviation “SPS” (it was still in our plots in the originally submitted version). Unfortunately, we found that this abbreviation is already in use for a different nonlinear process and we replaced it with “SPPS” in order to avoid confusion. The physical argument brought forward by the reviewer makes some sense, but ultimately we think it is of limited relevance and the other reviewers did not seem to mind. Especially because “SPS” is already in use, we would prefer to not adopt this change.

Reviewer:

Summary: since the damping of 2D plasmons is many orders of magnitude larger than the damping of sound waves, the feasibility of SPPS had to be studied and clearly demonstrated in much more details than it is done in the paper. Personally I do not believe that the SPPS phenomenon will ever work in structures with graphene or any other 2D materials and I did not find in the paper convincing arguments that the proposed effect can be observed indeed. The paper is very badly written: the authors carefully hide from the reader details of the theory rather than explain what and how they did. I do not recommend this manuscript for publication.

Response:

As we already pointed out: The reviewer has an incorrect idea of the propagation length of sound waves in SBS, which is actually very comparable to that of THz-plasmons in graphene.

Reviewers' comments:

Reviewer #3 (Remarks to the Author):

see report in the attached file

Comments of Reviewer 3 (italic font) and our responses

Reviewer:

In the 3rd version of the manuscript the authors provided some more details and explanations. However I still have doubts on the feasibility of the proposed effect and believe that this (very important) point should be more thoroughly and more critically analyzed in the manuscript. From the first version of the manuscript I had and still have impression that the authors too optimistically estimate the quality factor of PPs and hence overestimate the feasibility of the proposed idea.

Response:

We accept that the reviewer has this opinion and we don't think we could convince them otherwise.

Reviewer:

*1. The authors states that "THz graphene plasmons ... have quality factors between 10 and several 1000". The number of "several 1000" looks overly optimistic. This number is taken from calculations shown in Figure 2b, but this is only a theoretical value. Experimental quantities are much less impressive: e.g. in Nature Nanotech **6**, 630 (2011) the quality factor Q was about 1, in two papers of Basov and Koppens groups (Nature, vol **487** (2012)) it was about 10. In a recent paper Nature **557**, 530 (2018) the value of $Q \sim 130$ was achieved in graphene encapsulated between BN layers, i.e., in a structure where the influence of impurities is to a very large extent excluded, which means that further increase of the mobility is hard to expect. All these numbers are much less than the authors' theoretical "several 1000".*

Response:

The reviewer refers to the sentence "*Their propagation loss is in fact very comparable to that of GHz-range acoustic phonons found in SBS as illustrated by the fact that both THz graphene plasmons as well as GHz sound waves in technologically relevant materials [...] have quality factors between 10 and several 1000.*" In our plots we find plasmonic quality factors of at most 1500 in one of 4 curves. All other stay below 1000. We wanted to point out that acoustic and THz graphene plasmons are roughly in the same ballpark and wrote "several 1000" only because acoustic quality factors can reach 10000.

The reviewer cites Nature 557, 530 (2018), which is also what we used as our primary source for loss estimates. The reference measures losses at 27 THz and reports for $T = 60$ K an experimental quality factor of 130 and an intrinsic quality factor of > 970 . The reference

defines the quality factor as the ratio between real and imaginary parts of the wave number (which makes sense since the frequency is imposed by the laser). Our number for 27 THz at 60 K is 300, which we define as the ratio of real and imaginary parts of the frequency, which makes sense because the (real) wavenumber is imposed via the phase matching. The two definitions of the quality factor differ by the ratio of group and phase velocity of the plasmon at the operating point. This explains the discrepancy between our 300 and the reported 130. We note that our definition of the quality factor was mentioned in the manuscript.

At the risk of irritating readers from the SBS community, we dropped the “several” claim and reformulated the quote that all this hinges on. We replace

“Their propagation loss is in fact very comparable to that of GHz-range acoustic phonons found in SBS as illustrated by the fact that both THz graphene plasmons as well as GHz sound waves in technologically relevant materials [...] have quality factors between 10 and several 1000.”

with

“Their propagation loss is in fact very comparable to that of GHz-range acoustic phonons found in SBS as illustrated by the fact that both THz graphene plasmons as well as GHz sound waves in technologically relevant materials [...] have quality factors of orders of magnitude 10^1 – 10^3 .”

Reviewer:

2. Then, as seen from Figure 2b, Q factor becomes bigger than 1000 at Fermi energies bigger than 1 eV. I have already written in the previous report that the values of E_F larger than 0.4-0.5 eV are absolutely unrealistic. The authors agreed, but decided to leave these unrealistic numbers in the manuscript: “The reviewer is correct that electrostatic doping beyond 0.4-0.5 eV is increasingly difficult. However, recent discussions with peers have indicated that at least values up to 0.5/eV are not unrealistic. We decided to show data for higher energies anyway, because readers might find a way to reach them e.g. via chemical doping”. Well, maybe readers will find a way to reach 1 eV Fermi energy but do not use these extremely high numbers for justifying the feasibility of the effect.

Response:

The reviewer continues to suggest that we require doping levels beyond 0.5 eV, which is not the case. Our plots clearly show that the best absolute gains are predicted below 0.4 eV and the best relative gains just above 0.4 eV. Our plots show that there is no benefit in increasing the doping beyond that and we think that this information in itself is also worthwhile.

At no point did we concede that doping up to 0.5 eV was “absolutely unrealistic.” In fact, one of us co-authored a paper [Nano Lett. 15, 4393 (2015)] where electrically doping the inter-band threshold through the telecom band was used to modulate light. We know first-hand that it is possible to electrically tune graphene just beyond 0.5 eV. We did and still do agree that it is an engineering challenge to do so in this context.

We added a sentence to the end of the first paragraph of Section III: “Finally, we find that the overall performance declines above $E_F \approx 0.5$ eV despite the growing plasmonic quality factor. This is due to the resonant ponderomotive interaction and means that doping beyond the currently feasible levels would not provide any further benefit.”

Reviewer:

3. The authors hope that such a high values of Fermi energies may be achieved by chemical doping. But chemical doping means that a large number of impurities will be placed in the vicinity of graphene layer and this will strongly reduce the mobility of graphene electrons (e.g. Nat. Phys. 4, 377 (2008)). Thus, huge values of Q assumed by the authors turn out to be unachievable again.

Response:

We agree that chemical doping is likely to introduce impurities that reduce the quality factor. However, we do not hope for a miracle with chemical doping. As explained in our response to point 2, we know for a fact that electrical doping is sufficient to achieve the necessary levels of up to 0.5 eV, although certainly not trivial. We mention chemical doping in the manuscript and our previous response letter for completeness alongside other suggestions.

Reviewer:

4. I tried to check how the authors plotted the Stokes shift in Figure 2b. For the 2D plasmon spectrum I took the formula $\Omega = (e/\hbar)[2E_Fq/\kappa]^{1/2}$. For the wave vector q I took the value $q = 2k = 2\omega n/c$ (according to Fig 1a), where $n = 1.37$, as stated in the manuscript. As a result I got the curve shown below, where for κ (kappa) the value 1 was taken. Qualitatively it looks similar to the one shown in Figure 2b, but quantitatively the 2D plasmon frequency is by a factor of 1.5 smaller (60 THz at Fermi energy 2 eV vs. 90 THz in the manuscript). If my estimate is correct this should mean further reduction of the quality factor Q . If they are incorrect, how the curve in Figure 2b was obtained? It is not discussed in the manuscript.

Response:

We thank the reviewer for investing the effort of reproducing our plots in such detail. This is naturally a highly appreciated service to us. We agree with the general procedure outlined by the reviewer, except that they treat the narrow graphene ribbon as an infinitely extended graphene sheet. Like with acoustic modes and tubular microwave waveguides, lateral confinement increases the frequency for a given wave vector. The naive approach (that is exact e.g. in microwave waveguides) is to assume a sinusoidal mode profile. In our case, this simple approximation increases the frequency by about 50% and is within 3% of the analytical solution of a strip in vacuum [see Adv. Opt. Mater. 8, 1901473 (2020)]. We hope that this resolves the reviewer's issue.

In the previous rounds, we had taken the reviewer's questions regarding the dispersion relation of graphene with respect to the material itself and proximity effects such as acoustic plasmons. In this sense, we assumed the usual graphene dispersion relation (see also our response to point 7). However, given the previous discussion of silicon waveguides we should not have taken it for granted that the dispersion relation of a waveguide differs from that of the "bulk" material.

As an aside, we note that in order to provide a more conservative estimate for the opto-plasmonic coupling, we have always simply assumed a uniform lateral charge distribution for the plasmonic mode in our overlap integrals. In reality, the plasmonic mode profiles are strongly localized at the strip edges, improving the mode overlap. Therefore, we have underestimated the opto-plasmonic coupling in this aspect (and still do so), but we are only interested in ballpark numbers, anyway.

We added a sentence to paragraph 4 of Section II of the manuscript:

"We note that the dispersion relation of such ribbons differs from that of infinitely extended graphene due to the lateral confinement [34]."

Reviewer:

5. After my critique in the 2nd report the authors gave at last the numbers for the Drude relaxation rates (2.5 meV at 300K and 0.25 meV at 60 K) which they used for estimates of Q and took from Ref.[24]. But in [24] I read "Inspection of these data reveals a dramatic drop in the intrinsic plasmonic scattering rate from about 20 cm^{-1} at room temperature to $<2.0 \text{ cm}^{-1}$ at $T = 60 \text{ K}$ (we note the frequency unit conversion rule: $1 \text{ cm}^{-1} = 30 \text{ GHz} = 1.43 \text{ K}$ ". 20 cm^{-1} correspond to 3.5 meV and 2 cm^{-1} correspond to 0.35 meV. The difference between 2.5 meV and 3.5 meV is not too big, but it leads again to smaller values of $Q = \Omega/\gamma$. All these small inaccuracies, which seem to aim to artificially "increase" the quality

factor to a desired very high value, create a bad impression of the manuscript.

Response:

We thank the reviewer for catching this admittedly embarrassing mistake in converting the literature values. We corrected it and replotted the graphs in Fig. 2. The calculation itself was correct and overall there are no dramatic changes to the results or the message.

Reviewer:

*6. Further, in nonlinear experiments the radiation power is quite strong which may heat the 2D electron gas to a higher electron temperature T_e (for example, in the experiment Nat. Nanotech. **13**, 583 (2018) T_e was estimated to be higher than 1000 K). Since according to Ref. [24] the scattering rate increases by a factor of 10 when the temperature grows from 60 to 300 K, the heating of the electron gas by a strong optical power may lead to a further reduction of the quality factor.*

Response:

The reviewer is correct that most conventional nonlinear experiments use very large intensities, but that is not necessarily so in our case. The process mentioned by the reviewer certainly exists and we did ignore it in the previous versions of the manuscript, because in our opinion it did not relate to our proposal. It is an ultrafast transient heating of the electron gas upon absorbing a tightly focused femtosecond pulse. In contrast, we envisage pulses that are several orders of magnitude longer (as required by the plasmon frequency and quality factors) that are dissipated along the entire waveguide, which further reduces the local heating effect. Therefore, we had simply assumed thermal equilibrium between the different systems within the graphene sheet, specifically between the electron and the phonon gases.

Since we had not considered the possibility of unequal temperatures of the electron and phonon system and because of the potentially grave consequences suspected by the reviewer, we looked into this in depth and with an open mind, also seeking advice from other expert colleagues. We did find that our neglect was a bit premature and we thank the reviewer for sending us down that particular rabbit hole – it was quite interesting. However, we have to conclude that their comment is at least as premature and simplistic. We present our findings here in quite some detail.

First, we must understand the meaning of the electron temperature in the context of this problem. The reviewer suggests that an elevated electron gas temperature leads to drastically increased loss.

By conflating their reference with Ref. [24] they make the connection between the electron temperature and the temperature-dependence of the sheet conductivity. This is incorrect.

The intra-band dissipation in the sheet is due to inelastic scattering events between charge carriers and some other (quasi-)particles. Apart from static scattering sources such as impurity atoms, there are two candidates available as scattering partners: phonons and electrons. If the electron gas temperature had the effect suggested by the reviewer, inelastic carrier-carrier scattering would have to be at least comparable to carrier-phonon scattering. Furthermore, the rate of carrier-carrier scattering would have to dramatically increase with temperature. Regarding the first point, we note that 2d electron systems such as graphene are similar to 3d systems in that within the framework of Fermi liquid theory electron-electron collisions manifest as renormalized parameters of the Fermi quasi-particles, which constitute the non-interacting true charge carriers of the system. As a result, inelastic carrier-carrier scattering is a very weak process. Furthermore, the effect of temperature on the carrier population is fundamentally different to the phonon population due to the Fermionic nature of the former whereas the latter are collective Bosonic excitations with chemical potential zero. A change in temperature leads to a widening of the carrier distribution with an additional shift in the chemical potential (Fermi level) while maintaining the overall number of particles. As a result, the number of available scattering partners does not change with temperature although the number of available destination states does increase. In contrast, an increase in temperature leads to an increase of the total number of phonons to first approximation according to the Stefan-Boltzmann law, i.e. roughly with the third power of temperature, in addition to the aforementioned increase in the number of available destination states. This means it is unlikely that a high electron gas temperature would lead to significantly increased dissipation for low-energy electromagnetic waves. It does lead to a softening of the onset of inter-band transitions. In summary, we conclude that electron gas heating has no effect on the quality factors of plasmon polaritons in the intra-band regime, which we find for $E_F > 0.1$ eV. Here, the relevant temperature is the temperature of the lattice.

After having established that an elevated electron temperature does not lead to significantly increased Ohmic loss, we now must ask what kind of effects it will have. This means we now look for effects that are directly linked to the softening of the Fermi distribution rather than scattering events. The first consequence is increased dephasing of the electronic states leading to inhomogeneous broadening. As far as we are aware this is a minor effect compared to the homogeneous dissipative broadening of the resonance line and visible either at extremely high electron temperatures ($\gg 1000$ K) or at very low lattice temperatures. Probably the main effect of the electron temperature is the sharpness of the onset of the inter-band contribution to the

conductivity. This manifests in two ways. On the one hand, it leads to a softer transition between the regimes of low and high optical loss, which we take into account in our simulations. On the other hand, it impacts the maximally attainable gain figure, because one aspect of the increased gain at 60 K is due to a resonant ponderomotive nonlinearity at the inter-band threshold. This means that while the electron gas temperature does not influence the quality factor, it does affect the nonlinear coupling between the optical and plasmonic modes as well as the optical propagation length of the hybrid waveguide.

Given that the electron temperature will have an effect on our results, it is necessary to estimate a realistic temperature in our setting. We emphasize that it is very easy to underestimate the effect by inadvertently double-counting some of the mitigating effects listed below. Furthermore, it appears to us that the topic of hot electrons in ultrafast processes is somewhat divisive and we have no expertise of our own to make definite statements, but estimate the effect to the best of our knowledge. We feel that using the reviewer's reference as a starting point will lead to conservative estimates. They report an increase in temperature from room temperature to 1500 K (increase of 1200 K, with plenty of uncertainty) if inter-band transitions are possible and an increase of 200 K (from 300 to 500 K) if inter-band transitions are not permitted by the Fermi level. Since the best regime for observation of SPPS is just on the low-loss side of the inter-band threshold, the latter value lets the situation already look less dire. The electron gas temperature is determined by the energy dissipated over one electron-lattice relaxation constant, approximately 1 ps. Since the experiment used much shorter femto-second pulses, the entire energy absorbed from one pulse contributed to the electron gas heating. Assuming an absorption of 2.3% from a pulse fluence of 700 mJ/m^2 corresponds to an absorbed energy density of 15 mJ/m^2 in the reference in the inter-band regime. In contrast, we have pulses of at least 10 ps in mind, so we must compare to the product of the relaxation time and the *power* absorbed per unit area in our proposal.

In order to get an estimate for our proposal, we assume an optical decay length of 1 mm, which is a realistic number in the low-loss regime $E_F \geq 0.4 \text{ eV}$ at room temperature. Together with a strip width of 150 nm, we find an effective dissipated energy per electron-lattice relaxation time of $\approx 5 \text{ mJ/m}^2$ for each Watt of pump power. The resulting electron heating must be based on the reported 1200 K increase. As a result, we would estimate an electron gas heating in the ballpark of 300 K above the lattice temperature per Watt of pump power. This heating effect should drop by an order of magnitude at 60 K due to reduced optical absorption (we calculate waveguide loss as low as 1 dB/cm).

In the manuscript, we projected pump powers between 10 mW and 10 W. Clearly, the upper boundary becomes problematic due to this effect, but it appears the pulse powers below 300 mW are not unrealistic both at room temperature (leading to 100 K of heating) and at

60 K (leading to 10 K heating). Towards the lower end of projected pump power range, we do not anticipate any significant effect. In this regime, our initial assumption (thermal equilibrium between electron and phonon system) was perfectly fine.

In order to demonstrate the impact of thermal imbalance, we repeated the calculation with 60 K lattice temperature and 160 K electron gas temperature, which corresponds to 3 W pump power at that temperature and in the low optical loss regime. We find that the absolute gain drops by an order of magnitude and the relative gain by two orders of magnitude, but remains above the detection threshold. We included this calculation in Fig. 2 in the manuscript and added a brief discussion of the issue at the end of the first paragraph of Section III:

We note that for pump powers on the higher side, pump dissipation in the graphene causes a relevant difference between the temperatures of the electron gas and the phonon gas. This leads to reduced SPPS-gain and increased *optical* loss while leaving the polaritonic quality factor for the most part unchanged. The exact threshold for this effect depends strongly on optical pump power, (lattice) temperature, Fermi level and waveguide design (see the supplemental material for further discussion). For our configuration “1” at 60 K with $E_F \geq 0.4$ eV we estimate a temperature difference of 10 K at 300 mW pump power. To illustrate the effect, we show in Fig. 2 the impact of a very strong imbalance of 100 K expected for 3 W pump at 60 K. We note that this power level is already close to the destruction threshold of a silicon waveguide even with short picosecond pulses, so we present an extreme case.

Furthermore, we have added a lightly edited version of the above discussion as Section V to the supplementary material.

Reviewer:

7. In my previous report I raised a question about the 2D plasmon spectrum ($\omega^2 \sim q$ or $\omega^2 \sim q^2d$). The authors replied that they did not suggest using heavily doped semiconductors anywhere, but the question was different. In order to get high values of Q , they should have high values of Fermi energy. In order to get high values of Fermi energy one should use something like back or front gate and apply a large gate voltage to it. But having a conducting gate in the very vicinity of the 2D gas means that the frequency of 2D plasmons becomes smaller ($\omega^2 \sim q^2d$ instead of $\omega^2 \sim q$ where $qd \ll 1$). This would lead, again, to a reduction of the Q factor.

Response:

We thank the reviewer for clarifying their question. As the reviewer is perhaps aware, our group has played a leading role in the theoretical

exploration of the transition from the common regime with a square-root dispersion to the regime with acoustic-like dispersion, including possible nonlocal quantum effects emerging from the close proximity to the metal (see e.g. a recent PhD thesis from our group published by Springer–Nature, DOI: [10.1007/978-3-030-38291-9](https://doi.org/10.1007/978-3-030-38291-9), or Ref[33]). We are happy to reassure the reviewer that for any of the large distances d that we have in mind for the displacement of the gate electrode, the hybridisation of the graphene plasmons (with dispersion $\omega^2 \propto q$) and the surface-plasmon polaritons of the metallic gate is negligible.

Stimulated plasmon polariton scattering

C. Wolff*

Center for Nano Optics, University of Southern Denmark, Campusvej 55, DK-5230 Odense M, Denmark

N. A. Mortensen

Center for Nano Optics, University of Southern Denmark, Campusvej 55, DK-5230 Odense M, Denmark

Danish Institute for Advanced Study, University of Southern Denmark,

Campusvej 55, DK-5230 Odense M, Denmark and

Center for Nanostructured Graphene, Technical University of Denmark, DK-2800 Kongens Lyngby, Denmark

(Dated: May 19, 2020)

The plasmon and phonon polaritons of two-dimensional (2d) and van-der-Waals materials have recently gained substantial interest. Unfortunately, they are notoriously hard to observe in linear response because of their strong confinement, low frequency and longitudinal mode symmetry. Here, we propose a fundamentally new approach of harnessing nonlinear resonant scattering that we call *stimulated plasmon polariton scattering* (SPPS) in analogy to the opto-acoustic stimulated Brillouin scattering (SBS). We show that SPPS allows to excite, amplify and detect 2d plasmon and phonon polaritons all across the THz-range while requiring only optical components in the near-IR or visible range. We present a coupled-mode theory framework for SPPS and based on this find that SPPS power gains exceed the very top gains observed in on-chip SBS by at least an order of magnitude. This opens exciting new possibilities to fundamental studies of 2d materials and will help closing the THz gap in spectroscopy and information technology.

I. INTRODUCTION

The study of plasmon polariton excitations in two-dimensional (2d) materials [1] and the related class of van-der-Waals (vdW) materials [2] has recently gained considerable attention since they provide the means to very tightly confine, guide and manipulate electromagnetic fields from the few-THz range all the way into the mid-infrared. Furthermore and unlike conventional plasmonics based on metals, 2d material plasmon polaritons are highly sensitive to their electromagnetic, electronic and chemical environment, allowing for great tuning flexibility as well as suggesting their versatile use for sensing [3–5]. As an example, the extreme spatial light confinement paves the way to applications as varied as mid-infrared vibrational fingerprints of proteins [6], control of symmetry-forbidden atomic transitions [7], or single-photon nonlinear optics [8]. However, their greatest strengths — unrivaled mode confinement and operation in a hitherto poorly explored frequency regime — along with their peculiar mode symmetry also turn out to be a nuisance for experimental work. To date, the most promising avenue to overcome the extreme wave number mismatch between polaritons and free-space radiation is by scattering at discontinuities, e.g. introduced by the probe of a scanning near-field optical microscope (SNOM). or material discontinuities designed into the device [9]. As an alternative approach, the generation of graphene plasmon polaritons by difference frequency generation based on the intrinsic nonlinearities has been studied both theoretically [10, 11] and experimentally [12]. This has led to significant insights [13], but

is somewhat inefficient and does not seem very practical for applications beyond fundamental research. We suggest that this can be further drastically enhanced and harnessed for practical applications by borrowing ideas from the seemingly unrelated field of opto-mechanics, specifically of Brillouin scattering.

Stimulated Brillouin scattering (SBS) is the inelastic, resonant and self-amplifying back-scattering of light from a propagating acoustic wave in matter [14–16]. From its initial status as an academic curiosity, it has soon proven invaluable to characterize mechanical properties of materials at GHz frequencies — a difficult range for direct mechanical measuring techniques. More recently, it has attracted considerable attention for the realisation of flexible yet highly selective optical filters [17], novel light sources [18], the processing and buffering of optical signals [19, 20] and for the generation and amplification of coherent hypersonic waves e.g. following the concept of the so-called phonon laser [21].

We introduce the novel concept of inelastic, resonant and self-amplified scattering of light off propagating polaritons especially in 2d and vdW materials; a process that we refer to as *stimulated plasmon polariton scattering* (SPPS). In analogy to the conceptually similar SBS, this will not only allow for the detection of 2d plasmons at the most convenient optical wavelength, but also to accurately measure both the frequency and damping at the given wave number. In cases where the dispersion relation depends on a parameter (e.g. the Fermi energy E_F), the wide-band nature of SPPS allows to characterize this dependency over 1–2 orders of magnitude. In the case of graphene plasmon polaritons (GPPs), this means that all regimes of the dispersion relation (pure intra-band scattering, inter-band corrections and potentially even nonlocal effects [22–25]) can be experimentally char-

* cwo@mci.sdu.dk

FIG. 1. Illustration of the phase-matching in SPPS and schematics of two potential realisations.

(a) SPPS-interaction in the optical (straight solid) and polaritonic (curved solid) dispersion relation: The pump (green point, frequency ω and wave number k) is scattered into the counter-propagating Stokes mode (red point) where the difference in frequency and momentum (blue line) matches the polaritonic dispersion relation. This polariton mode (blue point, wave number $q \approx 2k$ for angular frequency $\Omega \ll \omega$) is amplified along the waveguide.

(b) Conceptual schematic of an SPPS-experiment in Stokes-seed configuration. A weak Stokes seed (red arrows) is injected at the rear end and amplified as it propagates towards the front end of a graphene-covered waveguide. This action is also illustrated by qualitative spectra where the heights of the colored peaks indicate to the relative amplitudes of the matching signals at the front and back of the waveguide. The bias voltage contact and the waveguide geometry shown here were intentionally kept oversimplified at this stage and in reality would require careful engineering. This configuration is best suited for the excitation of polaritons and for characterizing their dispersion relation, which can be tuned via the bias voltage contact. It is also the natural candidate for a first demonstration of SPPS as all input and output signals are in the optical domain and no injection or detection of THz waves is required.

(c) Conceptual schematic of an SPPS-experiment in polariton-seed configuration. A weak THz signal is injected through the bias contact (blue) and amplified along the waveguide. Simultaneously, an optical Stokes is generated (illustrated by qualitative spectra similar to panel b). This configuration is compelling because of the prospect to amplify or optically detect weak THz signals.

acterized in one setup. Beyond this use in fundamental science, one can expect to harness SPPS for use in optical components, but especially for the tunable narrow-band amplification and optical detection of signals in the THz range. In the remainder of this paper, we describe the principle of SPPS and conclude that it is experimentally observable in a standard silicon slot-waveguide covered with an appropriately biased graphene-monolayer.

The proposed mechanism bears similarities with the excitation of graphene plasmons [10] through difference-frequency generation (DFG) via the intrinsic Pockels nonlinearity of graphene [26], but our aim is different. Our focus is not the intrinsic graphene nonlinearity that unlocks DFG, but rather the consequences of this nonlinearity on the interplay between optical and polaritonic fields that are confined in a common long waveguide. We show that this will cause exponential gain for the THz polaritonic wave on a length scale well beyond its linear propagation length provided the nonlinear interaction is sufficiently strong. We find that among the nonlinear processes that can be found or introduced in a system composed of a dielectric waveguide combined with a graphene sheet, the ponderomotive nonlinearity [?] is sufficiently strong, but we stress that it is not necessarily the only possibility.

II. PRINCIPLE, THEORY AND FEASIBILITY

The SPPS process is conceptually related to the well-understood SBS process in nano-scale waveguides: in both cases a propagating low-frequency excitation with wave number q modifies the local permittivity of a waveguide through a nonlinear process. In SBS, the excitation is a sound wave and the nonlinear process is due to photoelasticity and the motion of the dielectric interface. In SPPS, the excitation is a localized polariton and the nonlinear interaction is either due to an intrinsic nonlinearity of the polariton system or to a Pockels nonlinearity in the waveguide. Thus, the polariton creates a travelling low-contrast grating in the waveguide, which can scatter an optical pump wave (with wave number k) into a counter-propagating Stokes wave if the difference in optical frequency and momentum matches the polaritonic dispersion relation (phase-matching). Assuming that the Stokes shift Ω is small compared to the optical pump frequency ω , the phase matching condition is approximately given by the ratio $q \approx 2k$ (see Fig. 1). In analogy to SBS, overall conservation of energy and momentum require that both the polariton field as the optical Stokes wave grow approximately exponentially along the waveguide. This process exists as soon as there is an efficient nonlinear coupling mechanism between the optical pump and the polariton. Naturally, this nonlinear coupling must be strong enough that the process is not

immediately quenched by the linear losses experienced by the plasmon polariton. At least for THz graphene plasmon polaritons this is actually not as bad as it might perhaps sound. Their propagation loss is in fact very comparable to that of GHz-range acoustic phonons found in SBS as illustrated by the fact that both THz graphene plasmons as well as GHz sound waves in technologically relevant materials [14, 16, 27–29] have quality factors of orders of magnitude 10^1 – 10^3 . Finally, neither SBS nor SPPs require long-range propagation of their respective non-optical excitations.

We now introduce the basic theoretical framework. In analogy to the theory of SBS [30], we describe the dynamics of the three participating waves within the framework of coupled-mode theory, where we assume a strict slowly varying envelope approximation. This means that the optical pump amplitude $a^{(p)}(y, t)$, the optical Stokes amplitude $a^{(s)}(y, t)$ and the plasmon polariton amplitude $b(y, t)$ all are assumed to vary slowly on the time scale of the slowest carrier in the system: the polariton frequency. This all leads to the nonlinearly coupled equations

$$\partial_y a^{(p)} + v_a^{-1} \partial_t a^{(p)} + \kappa_a a^{(p)} = -i\omega Q \mathcal{P}_a^{-1} a^{(s)} b^*, \quad (1a)$$

$$\partial_y a^{(s)} - v_a^{-1} \partial_t a^{(s)} - \kappa_a a^{(s)} = -i\omega Q^* \mathcal{P}_a^{-1} a^{(p)} b, \quad (1b)$$

$$\partial_y b + v_b^{-1} \partial_t b + \kappa_b b = -i\Omega Q \mathcal{P}_b^{-1} [a^{(p)}]^* a^{(s)}, \quad (1c)$$

where we assume the waveguide to extend along the y -direction, v_a is the group velocity of the optical mode and \mathcal{P}_a is the power to which it has been normalized, and κ_a is the optical decay parameter. Their plasmonic counterparts are v_b , \mathcal{P}_b , and κ_b , respectively. Natural choices for the normalization powers are $\mathcal{P}_a = \hbar\omega v_a/L$ and $\mathcal{P}_b = \hbar\Omega v_b/L$, respectively, with the unit length of waveguide $L = 1$ m. The interaction is mediated by the nonlinear overlap between the optical pump and Stokes modes and the permittivity change caused by the polaritonic mode:

$$Q = \langle \vec{e}^{(p)} | \Delta \varepsilon(\vec{e}^{(pol)}) | \vec{E}^{(s)} \rangle \\ = \int d^2 A \varepsilon_0 \sum_{ijk} [e_i^{(p)}]^* e_j^{(s)} e_k^{(pol)} \Delta \varepsilon_{ij}(\vec{e}^{(pol)}), \quad (2)$$

where the integral is carried out over the cross sectional plane of the waveguide, $\vec{e}^{(pol)}$ is the electric field distribution of the polariton mode and $\Delta \varepsilon_{ij}(\vec{e}^{(pol)})$ includes all suitable second-order nonlinearities in the system. This may include conventional Pockels nonlinearities $\chi^{(2)}$ in the waveguide or the polaritonic material or effects such as the ponderomotive nonlinearity, which we base our example on. In analogy to the theory of SBS [30] we can easily derive from Eqs. (1a–1c) the stationary power gain

$$G = \frac{2\omega\Omega|Q|^2}{\mathcal{P}_a^2 \mathcal{P}_b \kappa_b}, \quad (3)$$

which is the most interesting quantity in experiments. Besides inconsequential normalization constants and the frequencies ω , Ω , its main ingredients are the nonlinear

coupling integral Q and the polaritonic loss parameter κ_b . Therefore the reader concerned with loss in graphene might interpret G as a measure for how strong the non-linear coupling is compared to the linear polaritonic loss. This should be kept in mind when comparing the gain in successful SBS experiments to the numerical values that we calculate further below. The derivation of these equations and those underlying our numerical example is beyond the scope of the main text and provided as a supplementary note.

We will now apply this theoretical framework to an illustrative waveguide geometry that has not been optimized for the strongest possible gain, but rather selected for its simplicity. The main purpose of this example is to demonstrate that the concept is feasible, i. e. that sufficient gain can be achieved in a realistic setting. We focus entirely on the interplay of the opto-plasmonic nonlinearity and linear losses and leave important technological aspects of an actual experiment out. This includes especially the exact method of doping, which can be achieved e. g. electrostatically via a metal contact few microns above the structure, with direct electrical contacts [31] or by chemical doping. As a waveguide, we select a slot waveguide composed of two silicon beams with rectangular cross section (dimension 220×275 nm²) separated by a 150 nm air gap and operated at the standard telecom wavelength 1550 nm corresponding to a photon energy of 0.8 eV. We computed the electric field distribution of the fundamental mode (effective index 1.37) using the commercial finite element software COMSOL (see Fig. 2a). The slot waveguide was selected, because this configuration is known to support waveguide modes that are mostly confined in the gap between the silicon beams and are therefore an excellent starting point for nonlinear photonics in the silicon nanophotonics platform [32]. As a polaritonic system, we select a graphene monolayer that has been sandwiched between thin layers of hexagonal boron nitride to isolate it from adverse substrate effects. The polaritonic damping parameter κ_b depends on E_F . We calculate it from the Drude relaxation rates (3.5 meV at 300 K and 0.35 meV at 60 K) that were experimentally determined for this type of “vdW-sandwich” [24] and plot it implicitly in Fig. 2b as the polaritonic quality factor $\Omega/(2v_b\kappa_b)$. As mentioned before, this quality factor is the appropriate measure when comparing to loss found in SBS and other related scattering processes.

We consider two possible placements of the graphene: either as a narrow ribbon inside the waveguide gap for maximal field enhancement or placed on top of the waveguide as a more practical arrangement. We note that the dispersion relation of such ribbons differs from that of infinitely extended graphene due to the lateral confinement [33]. In our example system, the second order nonlinearity stems entirely from the intrinsic ponderomotive force of the electron system in graphene. It is a direct consequence of the dependence of the optical properties on the Fermi level and has been studied based on a pure intra-band model for the optical conductivity [34].

FIG. 2. (a) Sketch of the example waveguide and electric intensity distribution of the fundamental mode at 1550 nm. The red planes indicate the two considered placements of the graphene sheet.

(b) Total SPPS power gain, quality factor (i. e. polaritonic loss) and Stokes shift (i. e. polaritonic frequency) of the two example structures and at two temperatures (red dashed: $T = 300$ K; solid black: $T = 60$ K) as functions of the Fermi level E_F . (b) Total SPPS power gain, quality factor (i. e. polaritonic loss) and Stokes shift (i. e. polaritonic frequency) of the two example structures and at three different combinations of temperatures as functions of the Fermi level E_F . The temperatures are: $T = 300$ K, thermal equilibrium between phonon and electron gases (blue dash-dotted); $T = 60$ K, thermal equilibrium (black solid); red dashed: lattice at $T_{\text{latt}} = 60$ K and electron gas at $T_{\text{el}} = 160$ K (red dashed) as a result of high pump powers (see discussion in main text and supplemental material). The pink and green shaded areas indicate whether the optical pump can or cannot drive inter-band transitions, leading to a high-loss and a low-loss regime for the pump and Stokes signals. At the transition, the resonant inter-band contribution to the ponderomotive force creates a pronounced peak in the gain. (c) Relative SPPS gain figure $F = G/\kappa_a$ as a function of the Fermi level.

In addition, we also consider the inter-band contributions and finite-temperature corrections to the ponderomotive force that we have published elsewhere [35]. We should stress that while the ponderomotive interaction constitutes a second-order nonlinearity, it must not be confused with the second-order nonlinear electric susceptibility $\chi^{(2)}$. The latter is a third-rank tensor relating three electric field vectors, and the former is the sensitivity of the conductance tensor to variations in the scalar carrier density and therefore in general described by only a second-rank tensor.

In Fig 2b, we show the calculated SPPS-gain, quality factor and Stokes shifts of our example system for either sheet placement (left column: inside the gap, right column: on top of the waveguide) and at a moderately low temperature of 60 K as well as at room temperature. We find Stokes shifts throughout the entire THz-range, and for low Fermi-levels we predict SPPS-gains

in excess of 10^4 (Wm) $^{-1}$ over a fairly large parameter range with peaks approaching 10^6 (Wm) $^{-1}$. This has to be compared [36] to $1 - 10$ (Wm) $^{-1}$ for SBS in optical fibres [37, 38], $100 - 1000$ (Wm) $^{-1}$ in chalcogenide rib waveguides [16, 27] and $1000 - 10000$ (Wm) $^{-1}$ in well-engineered silicon waveguides [28, 29, 39]. This demonstrates that SPPS can be expected to provide levels of gain that are very competitive with similar nonlinear processes such as SBS even though the graphene sheet introduces linear loss, which can limit the useful waveguide length to below 1 mm.

III. DISCUSSION AND OUTLOOK

The example system of Fig. 2 was selected for its simplicity, maintaining conservative numbers for the aspect

FIG. 3. Sketches of two possibilities to boost the SPPS response via polaritonic dispersion engineering: (a) A ring-topology for the SPPS-active element side-coupled to a bus waveguide increases the gain dramatically, but requires the Stokes shift Ω to align with a multiple of the free spectral range (FSR) of the ring, significantly restricting any tunability. (b) A corrugation at $1/4$ the pump wavelength in the waveguide introduces a “slow-light” regime with corresponding Purcell enhancement for the polariton mode (solid blue lines compared to dashed for the uncorrugated waveguide). As the enhancement appears at a fixed wave number, the *polaritonic* Purcell enhancement is controlled by the *pump* frequency and can be achieved virtually irrespective of the actual Stokes shift Ω , which remains tunable e. g. via the Fermi level.

ratios and feature and gap sizes. Besides the technical problem of transferring a graphene-sandwich, the main experimental challenge is the linear loss introduced by the graphene. For a given photon energy (e. g. 0.8 eV as chosen in our example) and variable Fermi level, this introduces two quite distinct regimes: If the Fermi level is below half the photon energy (here: 0.4 eV), graphene is extremely lossy due to inter-band transitions, leading to high linear waveguide loss (in our example reaching power loss values up to 10000 dB/cm). For Fermi levels above half the photon energy, however, the loss drops by several orders of magnitude (we calculate values as low as 1 dB/cm at 60 K and just above the inter-band threshold). Since the linear optical loss limits the useful waveguide length, a natural figure of merit is the ratio between gain and loss:

$$F = G/\kappa_a, \quad (4)$$

which takes units of W^{-1} and whose inverse $1/F$ is the pump power level necessary to achieve net gain (the effect can remain detectable at considerably lower pump levels e. g. by pump modulation and lock-in amplification of the output Stokes). We plot this in Fig. 2c. For most Fermi levels, we find values around $0.1 \dots 1\text{ W}^{-1}$, which is enhanced up to 100 W^{-1} due to a ponderomotive resonance at the inter-band threshold. This means that peak pump powers from 10 W down to 10 mW would provide net-gain (the latter at liquid nitrogen temperatures and for waveguide lengths in the low millimeter range). This is very feasible with state-of-the-art silicon photonics techniques and sub-ns pulsed light sources would prevent damage due to excessive dissipation in the graphene and free carrier absorption in silicon. We note that for pump powers on the higher side, pump dissipation in the graphene causes a relevant difference between the temperatures of the electron gas and the phonon gas. This leads to reduced SPPS-gain and increased optical loss while leaving the polaritonic quality factor for the

most part unchanged. The exact threshold for this effect depends strongly on optical pump power, (lattice) temperature, Fermi level and waveguide design (see the supplemental material for further discussion). For our configuration “1” at 60 K with $E_F \geq 0.4\text{ eV}$ we estimate a temperature difference of 10 K at 300 mW pump power. To illustrate the effect, we show in Fig. 2 also the impact of a very strong imbalance of 100 K expected for 3 W pump at 60 K . We note that this power level is already close to the destruction threshold of a silicon waveguide even with short picosecond pulses, so we present an extreme case. Finally, we find that the overall performance declines above $E_F \approx 0.5\text{ eV}$ despite in continuously increasing plasmonic quality factor. This is due to the resonant ponderomotive interaction and means that doping beyond the currently feasible levels would not provide any further benefit.

We will now present some guidelines for the design of a more sophisticated geometry with superior performance over our simple slot-waveguide example. Firstly, we point out that the scaling between the polariton wavenumber q and the quality factor $\Omega/(2v_b\kappa_b)$ is counterintuitive to people from the opto-mechanics community. In SBS, acoustic loss grows super-linearly as q is increased, leading to a decrease in the quality factor. As a result, SBS-active elements are ideally designed to have a low optical mode index and high acoustic mode index. Curiously, we find the opposite effect in SPPS: the polaritonic quality factor *increases* as the wave number and frequency are increased. We believe this to be a direct result of decreasing polaritonic mode confinement. As a result, high-mode-index optical waveguide are more viable and especially the use of plasmonic waveguides based on noble metals appears quite interesting. Even the small propagation lengths are not necessarily a problem, because of the high gain and the inherent high optical loss of materials such as graphene in the inter-band regime. Furthermore, we point out that any form of field enhance-

ment will improve the relative gain figure F . While it is true that placing the polaritonic sheet in higher field enhancements inevitably increases the linear loss due to inter-band transitions, the SPPS process (like SBS and Raman scattering) is effectively a third-order nonlinearity and thus is bound to over-compensate the increase in loss.

Secondly, we find that dielectric loading of the polaritonic mode is quite beneficial. This is the reason why a graphene sheet positioned above a slot waveguide provides gains that are comparable to that of a narrow ribbon squeezed inside the gap, even though it is subject to considerably lower intensity enhancement and overall poorer mode overlap. It is not clear how screening of the polaritonic mode (e.g. due to the use of gold-plasmonic waveguides) affects the overall gain.

Finally, we anticipate that the gain can be dramatically boosted by dispersion engineering of both the optical and the polaritonic mode. One example for such a system is to shape the SPPS-active element into a ring that is side-coupled to a bus waveguide (Fig. 3a). This topology has been studied thoroughly in the context of SBS and can dramatically increase the overall gain, even allowing for spontaneous oscillation (lasing) [40]. However, it requires careful choice of the coupling parameter and close matching between the Stokes shift and the ring's free spectral range. As a result, it is a well understood and easily implementable concept, which however negates the opportunities offered by tuning the Fermi level. A second possibility is to corrugate the waveguide to introduce a band edge with associated slow-light regime. This is known to enhance SBS in slow-light fibers [41]. Alternatively, it is also possible to create a polaritonic band edge (Fig. 3b), reducing the polaritonic group velocity and hence mode power \mathcal{P}_b that appears in the denominator of Eq. (3). Despite the resonant nature of such a "slow-light" regime, this Purcell enhancement would be in fact *broad-band*. This may seem counter-intuitive at first, but is just a result of the fact that the band edge is always positioned at the Brillouin zone border, i.e. for a fixed wave number. Since $q \approx 2k$ is effectively fixed by the optical pump wave and Ω follows e.g. as a function of the Fermi level, a corrugation with a period of $1/4$ of the pump wave length in the waveguide will *always* introduce a Purcell enhancement irrespective of the value of Ω .

Beyond the value of SPPS as a tool for the fundamental science of atomically thin materials, we also anticipate potential practical applications once geometries with optimized gain have been developed. The first, obvious possibility is to adapt some of the current applications of SBS such as integrated light sources (similar to Raman-lasers), optical signal filtering and especially sensing, which is one of the areas where SBS is currently commercially applied. Like its acoustic counterpart, the polaritonic dispersion relation is highly sensitive to variations in the surrounding material. While the current SBS sensors mainly detect variations of the cladding's

acoustic impedance and static strain fields, similar SPPS sensors involving a sheet of 2d material would be highly sensitive to very thin adsorbed layers either through a change of permittivity, a reduction in the carrier mobility or shifts in the Fermi level [4]. Furthermore, the principles of Brillouin optical correlation domain analysis (BOCDA) could be adapted to pinpoint the perturbation along an SPPS sensor with sub-mm resolution [42].

Finally, we would like to emphasize another aspect that makes SPPS interesting for practical applications: the amplification and detection of signals in a narrow frequency band that can be selected anywhere in the THz-to mid-IR band. Like SBS, SPPS is a self-amplifying process that increases the amplitude of both the low-frequency excitation (sound in the case of SBS, polaritons in SPPS) as well as the optical Stokes signal exponentially along the waveguide. This means that a weak THz signal injected into a graphene sheet at the back of an SPPS-active waveguide could be detected in the optical domain as the Stokes signal. Additionally, the amplified THz-signal could be picked off at the front of the waveguide. Both the detection and amplification would be restricted to the narrow frequency window given by the plasmon frequency and quality factor at the given wave number and could be tuned within a wide range via the Fermi level. Such a tunable narrow-band detector in conjunction with a wide-band source (e.g. a thermal emitter) would have the same versatility as a narrow-band source in conjunction with a broad-band detector, and could be of considerable value to spectroscopy in the far-IR and THz regime.

In summary, we described the optical coupling to (e.g. plasmonic) polaritons in 2d and vdW materials through a new physical process that we call stimulated plasmon polariton scattering. We outlined the theoretical framework, and based on experimentally verified material parameters we showed that the process can be observed in an experimentally straight-forward system at moderate temperatures despite significant optical loss and that even net gain for a Stokes signal can be achieved with a continuous-wave pump in this example at the appropriate Fermi level.

ACKNOWLEDGMENTS

C. W. acknowledges funding from a MULTIPLY fellowship under the Marie Skłodowska-Curie COFUND Action (grant agreement No. 713694). The Center for Nano Optics is financially supported by the University of Southern Denmark (SDU 2020 funding). N. A. M. is a VILLUM Investigator supported by VILLUM Fonden (grant No. 16498). The Center for Nanostructured Graphene is sponsored by the Danish National Research Foundation (Project No. DNRFF103). We are deeply indebted to P.A. Goncalvez, Joel Cox, and especially C. Tserkezis for valuable comments on the manuscript.

AUTHOR CONTRIBUTIONS

C. W. conceived and conducted the research, both authors discussed the results and contributed to the manuscript.

CODE AVAILABILITY

The main conclusions do not rely on any bespoke algorithms or computer codes.

DATA AVAILABILITY

The main conclusions of this paper do not rely on any datasets.

-
- [1] T. Low, A. Chaves, J. D. Caldwell, A. Kumar, N. X. Fang, P. Avouris, T. F. Heinz, F. Guinea, L. Martin-Moreno, and F. Koppens, *Nat. Mater.* **16**, 182 (2017).
- [2] D. N. Basov, M. M. Fogler, and F. J. García de Abajo, *Science* **354**, aag1992 (2016).
- [3] F. J. García de Abajo, *ACS Photonics* **1**, 135 (2014).
- [4] S. Xiao, X. Zhu, B.-H. Li, and N. A. Mortensen, *Front. Phys.* **11**, 117801 (2016).
- [5] P. A. D. Gonçalves and N. M. R. Peres, *An introduction to graphene plasmonics*, 1st ed. (World Scientific, 2016).
- [6] D. Rodrigo, O. Limaj, D. Janner, D. Etezadi, F. J. García de Abajo, V. Pruneri, and H. Altug, *Science* **349**, 165 (2015).
- [7] N. Rivera, I. Kaminer, B. Zhen, J. D. Joannopoulos, and M. Soljačić, *Science* **353**, 263 (2016).
- [8] M. Gullans, D. E. Chang, F. H. L. Koppens, F. J. G. de Abajo, and M. D. Lukin, *Phys. Rev. Lett.* **111**, 247401 (2013).
- [9] M. Autore and R. Hillenbrand, *Nat. Nanotechnol.* **14**, 308 (2019).
- [10] X. Yao, M. Tokman, and A. Belyanin, *Phys. Rev. Lett.* **112**, 055501 (2014).
- [11] H. Rostami, M. I. Katsnelson, and M. Polini, *Phys. Rev. B* **95**, 035416 (2017).
- [12] T. J. Constant, S. M. Hornett, D. E. Chang, and E. Hendry, *Nat. Phys.* **12**, 124 (2015).
- [13] Z. Fei, G.-X. Ni, B.-Y. Jiang, M. M. Fogler, and D. N. Basov, *ACS Photonics* **4**, 2971 (2017).
- [14] R. W. Boyd, *Nonlinear optics, 3rd edition* (Academic Press, 2003).
- [15] R. W. Boyd, K. Rzaewski, and P. Narum, *Phys. Rev. A* **42**, 5514 (1990).
- [16] B. J. Eggleton, C. G. Poulton, and R. Pant, *Adv. Opt. Photonics* **5**, 536 (2013).
- [17] A. Choudhary, Y. Liu, B. Morrison, K. Vu, D.-Y. Choi, P. Ma, S. Madden, D. Marpaung, and B. J. Eggleton, *Sci. Rep.* **7**, 5932 (2017).
- [18] N. T. Otterstrom, R. O. Behunin, E. A. Kittlaus, Z. Wang, and P. T. Rakich, *Science* **360**, 1113 (2018).
- [19] H. Shin, J. A. Cox, R. Jarecki, A. Starbuck, Z. Wang, and P. T. Rakich, *Nat. Commun.* **6**, 6427 (2015).
- [20] B. Stillier, M. Merklein, C. Wolff, K. Vu, P. Ma, C. G. Poulton, S. J. Madden, and B. J. Eggleton, *Opt. Lett.* **43**, 4321 (2018).
- [21] K. Vahala, M. Herrmann, S. Knünz, V. Batteiger, G. Saathoff, T. W. Hänsch, and T. Udem, *Nat. Phys.* **5**, 682 (2009).
- [22] A. Woessner, M. B. Lundeberg, Y. Gao, A. Principi, P. Alonso-González, M. Carrega, K. Watanabe, T. Taniguchi, G. Vignale, M. Polini, J. Hone, R. Hillenbrand, and F. H. L. Koppens, *Nat. Mater.* **14**, 421 (2014).
- [23] M. B. Lundeberg, Y. Gao, R. Asgari, C. Tan, B. Van Duppen, M. Autore, P. Alonso-Gonzalez, A. Woessner, K. Watanabe, T. Taniguchi, R. Hillenbrand, J. Hone, M. Polini, and F. H. L. Koppens, *Science* **35**, 187 (2017).
- [24] G. X. Ni, A. S. McLeod, Z. Sun, L. Wang, L. Xiong, K. W. Post, S. S. Sunku, B.-Y. Jiang, J. Hone, C. R. Dean, M. M. Fogler, and D. N. Basov, *Nature* **557**, 530 (2018).
- [25] E. J. C. Dias, D. A. Iranzo, P. A. D. Gonçalves, Y. Hajati, Y. V. Bludov, A.-P. Jauho, N. A. Mortensen, F. H. L. Koppens, and N. M. R. Peres, *Phys. Rev. B* **97**, 245405 (2018).
- [26] J. L. Cheng, N. Vermeulen, and J. E. Sipe, *Scientific Reports* **7**, 43843 (2017).
- [27] R. Pant, C. G. Poulton, D.-Y. Choi, H. Mcfarlane, S. Hile, E. Li, L. Thevenaz, B. Luther-Davies, S. J. Madden, and B. J. Eggleton, *Opt. Express* **19**, 8285 (2011).
- [28] R. Van Laer, B. Kuyken, D. Van Thourhout, and R. Baets, *Nat. Photon.* **9**, 199 (2015).
- [29] R. Van Laer, A. Bazin, B. Kuyken, R. Baets, and D. Van Thourhout, *New J. Phys.* **17**, 115005 (2015).
- [30] C. Wolff, M. J. Steel, B. J. Eggleton, and C. G. Poulton, *Phys. Rev. A* **92**, 013836 (2015).
- [31] Y. Gind, X. Zhu, S. Xiao, H. Hu, F. L. Hagedorn, N. A. Mortensen, and K. Yvind, *Nano Lett.* **15**, 4393 (2015).
- [32] C. Koos, P. Vorreau, T. Vallaitis, P. Dumon, W. Bogaerts, R. Baets, B. Esembeson, I. Biaggio, T. Michinobu, F. Diederich, W. Freude, and J. Leuthold, *Nature Photon.* **3**, 216 (2009).
- [33] P. A. D. Gonçalves, N. Stenger, J. D. Cox, N. A. Mortensen, and S. Xiao, *Advanced Optical Materials* **8**, 1901473 (2020).
- [34] S. A. Mikhailov, *Phys. Rev. B* **84**, 045432 (2011).
- [35] C. Wolff, C. Tserkezis, and N. A. Mortensen, *New J. Phys.* **21**, 073046 (2019).
- [36] In order to avoid confusion we note that SBS-gains for bulk materials are often specified in alternate units of m/W. Gains specified in $(\text{Wm})^{-1}$ are more suited to nano-scale waveguides and differ from the former by the effective waveguide mode area.

- [37] M. Nikles, L. Thevenaz, and P. A. Robert, *J. Lightwave Technol.* **15**, 1842 (1997).
- [38] M. S. Kang, A. Nazarkin, A. Brenn, and P. S. J. Russell, *Nat. Phys.* **5**, 276 (2009).
- [39] E. A. Kittlaus, H. Shin, and P. T. Rakich, *Nat. Photon.* **10**, 463 (2016).
- [40] S. R. Mirnaziry, C. Wolff, M. J. Steel, B. J. Eggleton, and C. G. Poulton, *J. Opt. Soc. Am. B* **34**, 937 (2017).
- [41] M. Merklein, I. V. Kabakova, T. F. S. Büttner, D.-Y. Choi, B. Luther-Davies, S. J. Madden, and B. J. Eggleton, *Nat. Commun.* **6**, 6396 (2015).
- [42] A. Zarifi, B. Stiller, M. Merklein, Y. Liu, B. Morrison, A. Casas-Bedoya, G. Ren, T. G. Nguyen, K. Vu, D.-Y. Choi, A. Mitchell, S. J. Madden, and B. J. Eggleton, *J. Opt. Soc. Am. B* **36**, 146 (2019).

Supplementary Note 1 to “Stimulated plasmon polariton scattering”

C. Wolff and N. A. Mortensen

I. INTRODUCTION & PRELIMINARIES

In this supplementary note, we present the theoretical foundation for the paper “Stimulated Plasmon Polariton Scattering”. This is mainly a derivation of the coupled mode equations Eq. (1a-c) of the main paper and the expressions for the nonlinear coupling and total gain. This part follows in broad strokes an earlier paper of ours on the theory of Brillouin scattering [1]. Furthermore, we discuss important aspects such as the normalization of the polaritonic modes.

We consider an optical waveguide along the y -direction of a Cartesian system. It is composed of high-index materials as described by the permittivity $\varepsilon(x, z)$ in the transverse plane and optionally a Pockels-nonlinearity $\chi^{(2)}(x, z)$. We assume that the waveguide supports bound propagating modes. Furthermore, we assume that waveguide modes interact with a nearby material that supports low-frequency plasmon-polariton modes. In the following we will assume this to be graphene, but other materials from the wider class of van-der-Waals materials or systems such as a 2d electron gas confined in a quantum well structure are also promising candidates, but might require minor modifications to the formalism. We assume that the plasma frequency in the polaritonic material can be tuned. In the case of graphene this is via the Fermi energy E_F , e. g. through adsorption of surface dopants or via an external gate electrode.

In the following, we will use upper case letters \mathbf{E} and \mathbf{H} for physically observable (real-valued) electromagnetic fields. Corresponding modal quantities \mathbf{e} , \mathbf{h} are written in lower case. In order to simplify notation, we employ the Einstein summation convention where advantageous.

A. Optical modes

Inside the waveguide, we assume a forward-propagating pump wave with angular frequency ω and wave number k as well as a slightly red-shifted counter-propagating Stokes wave. We assume that the propagation loss of the optical modes is weak enough to be captured via perturbation theory within a coupled mode framework based on the solutions to the lossless optical wave equations

$$\nabla \times \nabla \times \mathbf{E} = -\mu_0 \partial_t^2 \mathbf{D}; \quad (1)$$

where \mathbf{E} and $\mathbf{D} = \varepsilon \mathbf{E}$ are the electric field and the electric displacement, $\varepsilon(\mathbf{r})$ is the permittivity distribution and all materials are assumed to have the vacuum permeability μ_0 . As basis functions for the subsequent coupled mode theory, we neglect the imaginary part of ε and solve for

stationary modes

$$\mathbf{E}^{(p)}(\mathbf{r}, t) \approx a^{(p)}(y) \mathbf{e}^{(p)}(x, z) \exp(iky - i\omega t) + \text{c.c.}, \quad (2)$$

$$\mathbf{E}^{(s)}(\mathbf{r}, t) \approx a^{(s)}(y) \mathbf{e}^{(s)}(x, z) \exp(-iky - i\omega t) + \text{c.c.}, \quad (3)$$

where $\mathbf{e}^{(p)}$ and $\mathbf{e}^{(s)}$ are the modal profiles of the forward-propagating pump and backward-propagating Stokes wave, respectively, and $a^{(p)}$ and $a^{(s)}$ are the corresponding mode amplitudes. For any but the simplest geometries the mode distributions $\mathbf{e}^{(p/s)}(x, z)$ must be calculated numerically e. g. using finite elements.

Since we are investigating a nonlinear process, the relevant equations are not independent of amplitude levels. Therefore, the optical modes must be normalized, e. g. with respect to the energy density

$$\mathcal{E}_a = 2 \int d^2r \varepsilon [\mathbf{e}^{(p)}]^* \cdot \mathbf{e}^{(p)} \quad (4)$$

$$= 2 \int d^2r \varepsilon [\mathbf{e}^{(s)}]^* \cdot \mathbf{e}^{(s)}, \quad (5)$$

where \mathcal{E}_a is the unit of energy per unit length of waveguide, whose value is in principle arbitrary; typical choices are based on energies of one Joule or one optical quantum $\hbar\omega$. The energy density is connected to the total power flux \mathcal{P}_a via the group velocity v_a :

$$\mathcal{P}_a = v_a \mathcal{E}_a. \quad (6)$$

B. Polaritonic modes

In analogy to the optical modes, we now describe the plasmon polaritons inside the graphene sheet in terms of modes based on similar assumptions as for the optical waveguide. We assume that the sheet supports a tightly localized polariton mode at angular frequency Ω and wave number q . In the regime of strong confinement, the field distribution of the polariton of a sheet that is situated at an interface between two dielectrics with permittivities ε_1 and ε_2 can be approximated as:

$$\mathbf{E}^{(\text{pol})}(\mathbf{r}, t) \approx b(y) \mathbf{e}^{(\text{pol})}(x, z) \exp(iqz - i\Omega t) + \text{c.c.}, \quad (7)$$

$$\mathbf{e}^{(\text{pol})}(x, z) \approx \begin{pmatrix} 0 \\ \pm 1 \\ i \end{pmatrix} E_0 \exp[-q|z|], \quad (8)$$

where E_0 is the longitudinal electric field amplitude at the sheet. The plus and minus signs apply above and below the sheet, respectively; the total discontinuity of the normal electric field component is given as $\Delta E = 2E_0$.

The modal energy per unit length and unit width of the sheet material (required for normalization of the coupled mode equations) comprises three parts: the electromagnetic energy in the dielectrics surrounding the sheet, the electromagnetic energy inside the sheet (which is of similar order) and the non-electromagnetic energy of the electron system of the sheet:

$$\mathcal{E}_b = \underbrace{\frac{2\varepsilon_0(\varepsilon_1 + \varepsilon_2)E_0^2}{q}}_{\text{outside sheet}} + \underbrace{\frac{\Im\{\sigma(\Omega)\}E_0^2}{\Omega}}_{\text{inside sheet}} + \mathcal{E}_{\text{int}}, \quad (9)$$

While the first two terms are universal in as far as they only require knowledge of the polaritonic dispersion relation [which is equivalent to the knowledge of $\sigma(\Omega)$], the third term is highly specific for the particular choice of material. For illustration purposes, we calculate the non-electromagnetic correction for a graphene plasmon polariton due to the degeneracy pressure of the free carriers at zero temperature in the next section. At room temperature, we expect the main findings to remain qualitatively valid, but stress that the electromagnetic parts must be recalculated with the finite-temperature conductivity.

C. Graphene

Ignoring nonlocal effects inside the graphene and assuming zero temperature for simplicity, its electromagnetic properties are characterized by a dispersive sheet conductance:

$$\sigma(\Omega) = \sigma_1(\Omega) + \sigma_2(\Omega), \quad (10)$$

$$\sigma_1(\Omega) = \frac{ie^2 E_F}{\pi \hbar^2 (\Omega + i\gamma)}, \quad (11)$$

$$\sigma_2(\Omega) = \frac{e^2}{4\hbar^2} \left[\Theta(\hbar\Omega - 2E_F) + \frac{i}{\pi} \ln \left| \frac{\hbar\Omega - 2E_F}{\hbar\Omega + 2E_F} \right| \right], \quad (12)$$

where e is the electron charge, \hbar is the Planck constant and Θ is the Heaviside function. Here, σ_1 describes the contribution to the overall conductivity from intra-band effects of the free carriers, whereas σ_2 describes inter-band transitions inside the graphene sheet.

We now calculate the correction of the total polaritonic energy due to the degeneracy pressure of the electron gas. To this end, we assume that the carrier density is a harmonic variation around an equilibrium value defined by the mean Fermi energy:

$$n(t) = n_0[1 + \alpha \sin(\Omega t)], \quad (13)$$

where α is a small parameter. The 2d energy density associated with a carrier density is given via the chemical potential $\mu(n) = \hbar v_F \sqrt{\pi n}$:

$$\mathcal{E}_{\text{deg}}(t) = \int_0^{n(t)} dn' \mu(n') = \frac{2\hbar v_F \sqrt{\pi n(t)^3}}{3}. \quad (14)$$

The degeneracy correction to the modal energy is the average of this expression over one polaritonic cycle:

$$\mathcal{E}_{\text{deg}} = \frac{\Omega}{2\pi} \int_0^{2\pi/\Omega} dt \mathcal{E}_{\text{deg}}(t) \quad (15)$$

$$= \frac{2\hbar v_F \sqrt{\pi n_0^3}}{3} \left[1 + \frac{3\alpha^2}{16} + \mathcal{O}(\alpha^4) \right]. \quad (16)$$

$$= \mathcal{E}_{\text{deg}}^{(0)} + \mathcal{E}_{\text{int}} + \mathcal{O}(\alpha^4). \quad (17)$$

Here, $\mathcal{E}_{\text{deg}}^{(0)}$ is the energy introduced by doping, whereas the leading order correction with respect to α is taken as the modal energy of the polariton. By relating the variation αn_0 of the charge carrier density to the discontinuity of the electric normal field via Maxwell's divergence equation

$$\varepsilon_0(\varepsilon_1 + \varepsilon_2)E_0 = \alpha e n_0, \quad (18)$$

we can express the degeneracy correction in terms of the polaritonic field amplitude:

$$\mathcal{E}_{\text{int}} = \frac{\pi[\hbar v_F \varepsilon_0(\varepsilon_1 + \varepsilon_2)]^2}{8e^2 E_F} E_0^2. \quad (19)$$

We find for the total polaritonic energy per unit area of graphene:

$$\mathcal{E}_b = \left[\underbrace{\frac{4\varepsilon_0(\varepsilon_1 + \varepsilon_2)}{q}}_{\text{e.g. } 2.0 \times 10^{-16} \text{ F}} + \underbrace{\frac{2\Im\{\sigma(\Omega)\}}{\Omega}}_{\text{e.g. } 1.0 \times 10^{-16} \text{ F}} + \underbrace{\frac{2\pi[\hbar v_F \varepsilon_0(\varepsilon_1 + \varepsilon_2)]^2}{8e^2 E_F}}_{\text{e.g. } 2.8 \times 10^{-19} \text{ F}} \right] \frac{E_0^2}{2}, \quad (20)$$

where the prefactor takes on the meaning of an effective capacitance. Numerical values for the respective contributions have been evaluated for for the example $\varepsilon_1 = 1$, $\varepsilon_2 = 12$, $E_F = 0.1 \text{ eV}$, $\Omega = 0.1 E_F / \hbar = 1.5 \times 10^{13} \text{ s}^{-1}$, $q = 2.3 \times 10^6 \text{ m}^{-1}$. Clearly, this formula is reliable only in a low-loss regime. However, we can see that the in-sheet correction to the electromagnetic energy cannot be neglected, whereas the correction due to degeneracy pressure is (at least for this example) minimal.

II. MODAL EQUATIONS

We now derive nonlinear coupled mode equations for the stimulated plasmon polariton scattering problem.

A. Basic coupled mode equations

The optical equation of motion can be obtained from the wave equation:

$$\nabla \times \nabla \times \mathbf{E} = -\mu_0 \partial_t^2 \mathbf{D}. \quad (21)$$

To this end, we first assume that the electric field is the sum of the two optical modes

$$\mathbf{E} = a^{(p)}(y)\mathbf{e}^{(p)} \exp(iky - i\omega t) + a^{(s)}(y)\mathbf{e}^{(s)} \exp(-iky - i\omega t) + \text{c.c.} \quad (22)$$

In contrast, the electric displacement is the sum of the optical modes and a nonlinear polarization $\mathbf{P}^{(\text{NL})}$:

$$\mathbf{D} = \varepsilon a^{(p)}(y)\mathbf{e}^{(p)} \exp(iky - i\omega t) + \varepsilon a^{(s)}(y)\mathbf{e}^{(s)} \exp(-iky - i\omega t) + \text{c.c.} + \mathbf{P}^{(\text{NL})}. \quad (23)$$

We then insert this ansatz into the wave equation and project back onto either optical mode $\mathbf{e}^{(p/s)}$. We average over a time scale much larger than an optical cycle in order to isolate the dynamics of the slowly varying envelopes $a^{(p/s)}$ and we neglect second-order derivatives. We thus arrive at the optical equations of motion:

$$\partial_y a^{(p)} + v_a^{-1} \partial_t a^{(p)} + \kappa_a a^{(p)} = -i\omega \mathcal{P}_a^{-1} \langle \mathbf{e}^{(p)} | \mathbf{P}^{(\text{NL})} \rangle \quad (24)$$

$$\partial_y a^{(s)} - v_a^{-1} \partial_t a^{(s)} - \kappa_a a^{(s)} = -i\omega \mathcal{P}_a^{-1} \langle \mathbf{e}^{(s)} | \mathbf{P}^{(\text{NL})} \rangle \quad (25)$$

Here, \mathcal{P}_a appears from the overlap integral of the optical mode with itself and the source terms are the phase-matched projections of the nonlinear polarization onto the optical modes. The decay parameter κ_a is given as the overlap between the optical mode and the dielectric loss of the waveguide:

$$\kappa_a = \int d^2r \mathbf{e}^{*(p)} \Im\{\varepsilon\} \mathbf{e}^{(p)}. \quad (26)$$

In essentially the same way, the equation of motion is derived for the plasmon polariton envelope $b(y)$ from the optical wave equation in the THz range:

$$\partial_y b + v_b^{-1} \partial_t b + \kappa_b b = -i\Omega Q^* \mathcal{P}_b^{-1} [\langle \mathbf{e}^{(\text{pol})} | \mathbf{S}^{(E)} \rangle + \langle n^{(\text{pol})} | S^{(n)} \rangle], \quad (27)$$

where κ_b describes the absorption of the polaritonic mode along the sheet due to the real part of the sheet conductance $\sigma(\Omega)$ and $\mathbf{e}^{(\text{pol})}$ and $n^{(\text{pol})}$ are the electric field distribution and carrier density distribution of the polaritonic mode. The source terms $\mathbf{S}(\mathbf{r}, t)^{(E)}$ and $S(\mathbf{r}, t)^{(n)}$ to the polaritonic mode due to the optical pumping will be discussed in the next section.

The perturbative treatment of the losses is of course not as good an approximation for the polaritonic amplitude as it is for the optical ones, but should suffice for our purposes. This is based on the experience with the problem stimulated Brillouin scattering, where the lossless mode approximation is very successfully applied to the acoustic part of the problem, which features typical quality factors of several 100 to several 1000 just as we predict in the present work.

B. Nonlinear coupling

We assume that the coupling between the optical and polaritonic modes is due to three-wave mixing via some second order nonlinearity somewhere in the system. Therefore, we assume that the nonlinear polarization is the mixing product between the total optical field and the polaritonic field:

$$P_l^{(\text{NL})} = (\mathbf{E}_m^{(p)} + \mathbf{E}_m^{(s)}) (\varepsilon_0 \chi_{lmn}^{(2)} \mathbf{E}_n^{(\text{pol})} + \Pi_{lm} n^{(\text{pol})}), \quad (28)$$

where $\chi^{(2)}$ is the conventional second-order susceptibility e.g. due to the Pockels effect in the waveguide material and $n(\mathbf{r})$ is the local deviation of the carrier density from the average in the graphene sheet. The symbol Π describes the ponderomotive effect in graphene and only introduced here to keep the notation manageable and highlight the structure of the equations; in the next section we will not use it further and instead derive the ponderomotive expressions explicitly. Analogously, we assume that the polaritonic driving term is the mixing product between the two slightly detuned optical fields:

$$S_l^{(E)} = \chi_{lmn}^{(2)} \mathbf{E}_m^{(p)} \mathbf{E}_n^{(s)}, \quad (29)$$

$$S^{(n)} = \Pi_{mn} \mathbf{E}_m^{(p)} \mathbf{E}_n^{(s)}. \quad (30)$$

When evaluating the overlap products on the right hand sides of the coupled mode equations and performing the time averages, we find:

$$\langle \mathbf{e}^{(p)} | \mathbf{P}^{(\text{NL})} \rangle = a^{(s)} b \int d^2r [\chi_{lmn}^{(2)} e_l^{*(p)} e_m^{(s)} e_n^{(\text{pol})} + \Pi_{lm} e_l^{*(p)} e_m^{(s)} n^{(\text{pol})}], \quad (31)$$

$$\langle \mathbf{e}^{(s)} | \mathbf{P}^{(\text{NL})} \rangle = a^{(p)} b^* \int d^2r [\chi_{lmn}^{(2)} e_l^{*(s)} e_m^{(p)} e_n^{*(\text{pol})} + \Pi_{lm} e_l^{*(s)} e_m^{(p)} n^{*(\text{pol})}], \quad (32)$$

$$\langle \mathbf{e}^{(\text{pol})} | \mathbf{S} \rangle = a^{(p)} a^{*(s)} \int d^2r \chi_{lmn}^{(2)} e_l^{(\text{pol})} e_m^{*(s)} e_n^{(p)} \quad (33)$$

$$\langle n^{(\text{pol})} | \mathbf{S} \rangle = a^{(p)} a^{*(s)} \int d^2r \Pi_{lm} n^{*(\text{pol})} e_l^{*(s)} e_m^{(p)}. \quad (34)$$

By introducing the coupling integral

$$Q = \int d^2r [\chi_{lmn}^{(2)} e_l^{*(p)} e_m^{(s)} e_n^{(\text{pol})} + \Pi_{lm} e_l^{*(p)} e_m^{(s)} n^{(\text{pol})}], \quad (35)$$

we obtain the final coupled mode equations:

$$\partial_y a^{(p)} + v_a^{-1} \partial_t a^{(p)} + \kappa_a a^{(p)} = -i\omega Q \mathcal{P}_a^{-1} a^{(s)} b^*, \quad (36)$$

$$\partial_y a^{(s)} - v_a^{-1} \partial_t a^{(s)} - \kappa_a a^{(s)} = -i\omega Q^* \mathcal{P}_a^{-1} a^{(p)} b, \quad (37)$$

$$\partial_y b + v_b^{-1} \partial_t b + \kappa_b b = -i\Omega Q \mathcal{P}_b^{-1} [a^{(p)}]^* a^{(s)}, \quad (38)$$

The most natural source for a second order nonlinearity in the system is of course the Pockels effect. This can

be introduced, e. g. by composing the waveguide at least partially of a nonlinear material such as lithium niobate. We cannot say much more about such a setup except that we could not obtain an appreciable SPPS-gain based on this effect alone. The problem is that the polaritonic mode does not penetrate the waveguide enough to lead to a sufficient nonlinear mode overlap. We do, however, emphasize that this approach might prove very useful for the excitation of polaritons in materials other than graphene.

At least in the case of graphene the dominant second-order nonlinearity for the SPPS-process turns out to be the ponderomotive interaction, i. e. the local shift of the Fermi energy inside the graphene sheet as a result of the beat between the optical fields and – conversely – the emergence of a dynamic grating in the graphene conductance from a plasmon polariton.

C. Ponderomotive nonlinearity in graphene

We consider the case of a polariton in the THz-range and ask how this modulates the optical conductivity of graphene. This is a second-order nonlinearity, but not a conventional Pockels nonlinearity. The reason is that the polaritonic mode has even symmetry with respect to the sheet plane while the Pockels effect would describe the impact of a homogeneous electric field (which has odd symmetry) on the conductivity. Therefore (and unlike a Pockels nonlinearity) this particular type of second-order nonlinearity is not symmetry-forbidden in a system with inversion symmetry. One characteristic of materials like graphene is the very strong optical anisotropy; while the electrons are nearly free within the plane of the sheet, they can barely move in the normal direction. As a result, the ponderomotive effect arising from the plasma properties of the electron system interacts only with the in-plane components of the electric field.

The idea is that the polaritonic mode is in essence a fluctuation in carrier density, and therefore accompanied by a spatial modulation of the Fermi level E_F . This clearly implies the assumption that the polariton does not cause too much unrest in the electron system, but rather moves carriers around in a quasi-adiabatic way. The Fermi level in turn controls the intra-band conductance of graphene through

$$\sigma_1(\omega) = \frac{ie^2}{\pi\hbar^2(\omega + i\gamma)} \cdot E_F, \quad (39)$$

where ω is the optical angular frequency. The Fermi-level is also linked to the total carrier density n_{tot} :

$$E_F = \hbar v_F \sqrt{\pi n_{\text{tot}}} = \hbar v_F \sqrt{\pi(n_0 + n)}, \quad (40)$$

where n_0 is the equilibrium carrier density due to doping and n is the carrier density due to the polariton. The latter finally depends on the electric field discontinuity

across the sheet:

$$n = \frac{q\varepsilon_0(\varepsilon_1 + \varepsilon_2)E_0}{e}, \quad (41)$$

where q is the wave number of the polariton and doubles as the localization length of the electric field normal to the sheet. The nonlinearity in question (evaluated at $n = 0$) is then:

$$\frac{\partial\sigma_1}{\partial E_0} = \underbrace{\frac{ie^2}{\pi\hbar^2(\omega + i\gamma)}}_{(\partial\sigma_1)/(\partial E_F)} \cdot \underbrace{\frac{\hbar v_F \sqrt{\pi}}{2\sqrt{n_0}}}_{(\partial E_F)/(\partial n)} \cdot \underbrace{\frac{q\varepsilon_0(\varepsilon_1 + \varepsilon_2)}{e}}_{(\partial n)/(\partial E_0)}. \quad (42)$$

With the Fermi wave number $q_F = \sqrt{\pi n_0}$, this can be simplified:

$$\frac{\partial\sigma_1}{\partial E_0} = \frac{e\varepsilon_0 v_F}{2\hbar} \cdot \frac{i(\varepsilon_1 + \varepsilon_2)}{\omega + i\gamma} \cdot \frac{q}{q_F}. \quad (43)$$

It should be repeated that q and ω characterize different waves: q is the polaritonic wave number and ω is the optical angular frequency. In summary, we have found that a plasmon polariton in a graphene sheet is accompanied by an optical grating with the same period as the plasmon polariton. This dynamic grating can back-scattering light from the optical pump mode into the optical Stokes mode and thereby mediate stimulated plasmon polariton scattering.

The complementary effect (the source term to the polaritonic excitation due to the optical pumps) is derived from the same mathematical quantity $(\partial\sigma)/(\partial E_0)$, but with a different physical interpretation as a ponderomotive force in the 2d electron plasma. At this point we forgo an in-depth discussion of the connection between dynamic grating effect and the ponderomotive force via Onsager relations as well as the derivation of the inter-band contributions to this effect at finite temperature. Instead, we refer to our recent work [2] and only state the expressions that are relevant for the present work. The full optical conductivity of graphene at finite temperature has been calculated in Refs. [3, 4]

$$\begin{aligned} \frac{\sigma^{(T>0)}(\omega)}{\sigma_K} &= \frac{2iE_F}{\hbar(\omega + i\gamma)} + \frac{\pi}{4} \left(\tanh \frac{\hbar\omega + 2E_F}{4k_B T} \right. \\ &\quad \left. + \tanh \frac{\hbar\omega - 2E_F}{4k_B T} \right. \\ &\quad \left. + \frac{i}{\pi} \ln \frac{(\hbar\omega - 2E_F)^2 + (2k_B T)^2}{(\hbar\omega + 2E_F)^2} \right), \quad (44) \end{aligned}$$

where $\sigma_K = e^2/h \simeq 3.87 \times 10^{-5} \text{ S}$ is the inverse von-Klitzing constant, k_B is the Boltzmann constant. From this, we derived the ponderomotive nonlinearity

$$\begin{aligned} \frac{\partial\sigma^{(T>0)}}{\partial E_0} &= \frac{1}{2} \cdot \frac{\partial\sigma^{(1)}}{\partial E_0} \cdot \left[2 + \frac{\hbar\omega}{\hbar\omega + 2E_F} \right. \\ &\quad \left. + \frac{\hbar\omega(\hbar\omega - 2E_F)}{(\hbar\omega - 2E_F)^2 + (2k_B T)^2} \right], \quad (45) \end{aligned}$$

expressed in terms of the quantity given in Eq. (43) for simplicity. In this form, the square brackets express the fact that the inter-band threshold leads to a resonant enhancement of the ponderomotive force, which translates to a peak in the SPPS response as reported in the main text.

III. POWER GAIN AND GAIN SCALING

We now derive the expression presented in the main paper for the total SPPS-gain. To this end, we start with the coupled mode equations

$$\partial_y a^{(p)} + v_a^{-1} \partial_t a^{(p)} + \kappa_a a^{(p)} = -i\omega Q \mathcal{P}_a^{-1} a^{(s)} b^*, \quad (46)$$

$$\partial_y a^{(s)} - v_a^{-1} \partial_t a^{(s)} - \kappa_a a^{(s)} = -i\omega Q^* \mathcal{P}_a^{-1} a^{(p)} b, \quad (47)$$

$$\partial_y b + v_b^{-1} \partial_t b + \kappa_b b = -i\Omega Q \mathcal{P}_b^{-1} [a^{(p)}]^* a^{(s)}, \quad (48)$$

and assume both steady state and a local polaritonic response. The latter is justified if the plasmon polariton propagation length is smaller than the expected length scale on which the optical amplitudes change. This is justified in the case presented in the main paper. Due to the steady state assumption, all time derivatives vanish and the local polaritonic response assumption also removes the spatial derivative in the equations for $b(y)$. Therefore the polaritonic amplitude can be explicitly solved:

$$b = -i \frac{\Omega Q}{\mathcal{P}_b \kappa_b} [a^{(p)}]^* a^{(s)}. \quad (49)$$

By inserting this in the steady-state optical equations, we find:

$$\partial_y a^{(p)} + \kappa_a a^{(p)} = \frac{\omega \Omega |Q|^2}{\mathcal{P}_a \mathcal{P}_b \kappa_b} |a^{(s)}|^2 a^{(p)}, \quad (50)$$

$$\partial_y a^{(s)} - \kappa_a a^{(s)} = - \underbrace{\frac{\omega \Omega |Q|^2}{\mathcal{P}_a \mathcal{P}_b \kappa_b}}_{=\Gamma} |a^{(p)}|^2 a^{(s)}, \quad (51)$$

where Γ is the amplitude gain coefficient of the SPPS process and implicitly depends on the power unit \mathcal{P}_a used for normalizing the optical modes. In practice, most experiments measure the power gain, which relates optical power levels rather than arbitrarily normalized amplitudes. The power carried by the pump and Stokes modes are given by:

$$P^{(p/s)} = \mathcal{P}_a |a^{(p/s)}|^2. \quad (52)$$

Its y -derivative can be found via the product rule:

$$\partial_y P^{(p/s)} = \mathcal{P}_a a^{*(p/s)} \partial_y a^{(p/s)} + \text{c.c.} \quad (53)$$

By inserting this in Eqs. (50, 50), we find:

$$\partial_y P^{(p)} + \kappa_a P^{(p)} = \frac{2\omega \Omega |Q|^2}{\mathcal{P}_a^2 \mathcal{P}_b \kappa_b} P^{(s)} P^{(p)}, \quad (54)$$

$$\partial_y P^{(s)} - \kappa_a P^{(s)} = - \underbrace{\frac{2\omega \Omega |Q|^2}{\mathcal{P}_a^2 \mathcal{P}_b \kappa_b}}_{=G} P^{(p)} P^{(s)}, \quad (55)$$

where G finally is the SPPS power gain coefficient as presented in the main paper.

IV. COMPARISON WITH SBS

Confusion may arise as to the impact of polaritonic loss in SPPS. One first source of confusion can be the fact that the SPPS amplification effect stretches over a length orders of magnitude greater than the polariton propagation length. The short answer is that while the polariton may only propagate for 1-10 microns, the optical Stokes wave carries its information and together with the pump generates an amplified copy of the polaritonic signal further towards the pump. A second source of confusion can be the general perception that (especially plasmon) polaritons are very lossy while sound is nearly lossless and therefore SBS might seem massively more viable than SPPS. Here, the short answer is that acoustic loss increases dramatically with frequency and SBS involves GHz sound waves with a propagation length of 1-100 microns (depending on the material). We will now provide some discussion of how SBS and SPPS compare.

A. Comparison of loss figures

The appropriate measure for the loss of the non-optical field in both SBS and SPPS is the propagation length l . It can be easily found from the quality factor Q and the acoustic/plasmonic wavenumber q :

$$l = Q/q.$$

The quality factor in turn can be found either as the ratio of wavenumber and decay constant (just the inverse of the equation above) or the ratio of acoustic frequency and damping rate. From the phase matching diagram Fig.1a of the main text, which is identical for both SBS and SPPS, it is clear that $q \approx 2k$ and therefore very similar in both processes. This reduces the question of propagation lengths to comparing the quality factors Q of both processes. We find Q between 10 and 1000 in SPPS (see Fig.2b). As for the losses in SBS, Boyd [5] lists Stokes frequencies (denoted Ω_B) and damping rates (denoted Γ_B) in various bulk solids and liquids. By dividing Ω_B and Γ_B , one easily verifies that SBS has Q-factors in the same ballpark as SPPS. This also holds for material systems that have been the focus of recent SBS-research such as integrated soft glass waveguides ($\Omega/2\pi = 7.7$ GHz, $\Gamma/2\pi = 34$ MHz [6], corresponding to $Q = 226$) or silicon waveguides (e.g. $Q = 306$ in Ref. [7]). We find that SBS and SPPS suffer from comparable amounts of loss in the non-optical fields.

B. Competition between loss and nonlinear coupling

The viability of SPPS cannot be shown by comparing its typical linear loss figures with those of SBS. In addition, it is necessary to demonstrate that the nonlinear coupling is also strong enough to be comparable with SBS. The relevant parameter for such a comparison is the gain coefficient G . It actually already includes the loss in the form of the decay constant κ_B [see Eq. (55)]. As a result, G scales with the quality factor and therefore expresses precisely how strong the nonlinearity is compared to the loss of the non-optical wave (sound in SBS, plasmons in SPPS).

Therefore the question how the competition between nonlinear coupling and loss in SPPS compares to that in SBS can be decided by comparing the gain coefficients G . In SBS the gain is stated in one of two ways: As a material parameter (units m/W) appropriate for quasi-bulk systems such as fibres or as a waveguide parameter (units $(\text{Wm})^{-1}$) appropriate for complex nano-scale systems such as our example. The two gain figures are related by the effective mode area of the optical modes. Boyd lists the former quantity and based on an effective mode area of $0.1, \mu\text{m}^2$ (comparable to the mode area in our example), this translates to gains between $15 (\text{Wm})^{-1}$ for CS_2 and $\sim 4.5 \text{Wm}^{-1}$ for water and silica. A further discussion of this can be found in the main manuscript.

V. POWER HANDLING AND ELECTRON GAS HEATING IN GRAPHENE

Most conventional nonlinear experiments use very large intensities and observe strong heating of the electron gas. For the most part, it is an ultrafast transient effect upon absorbing a tightly focused femtosecond pulse. In contrast, we envisage pulses that are several orders of magnitude longer and that are dissipated along the entire waveguide, which further reduces the local heating effect. This requires a closer look to which extent electron gas heating might be detrimental to our proposal.

The dissipation in the sheet is due to inelastic scattering events between charge carriers and some other (quasi-)particles. Apart from static scattering sources such as impurities, there are mainly two candidates available as scattering partners: phonons and electrons. First, we note that 2d electron systems such as graphene are similar to 3d systems in that within the framework of Fermi liquid theory electron-electron interactions are absorbed in renormalized parameters for the Fermi quasi-particles, which constitute the non-interacting charge carriers of the system. As a result, inelastic carrier-carrier scattering is a very weak process to begin with. Furthermore, the effect of temperature on the carrier population is fundamentally different to the effect on the phonon population due to the Fermionic nature of the former whereas

the latter are collective Bosonic excitations with chemical potential zero.

A change in temperature leads to a widening of the carrier distribution with an additional shift in the chemical potential (Fermi level) while maintaining the overall number of particles. As a result, the number of available scattering partners does not change with temperature although the number of available destination states does increase. In contrast, the number of phonons increases with temperature to first approximation according to the Stefan–Boltzmann law (with the third power of temperature in 3d systems), in addition to the aforementioned increase in the number of available destination states. As a result, it is implausible that a high electron gas temperature leads to significantly increased dissipation for electromagnetic excitations in the intra-band regime. In summary, we conclude that electron gas heating has no significant effect on the quality factors of plasmon polaritons in the intra-band regime, which we find for $E_F > 0.1 \text{eV}$. In this regime the plasmon polariton loss is determined by the temperature of the lattice.

After having established that an elevated electron temperature does not lead to significantly increased Ohmic loss for the plasmon polaritons, we now address what kind of effects it will have, i.e. effects that are directly linked to the softening of the Fermi distribution rather than scattering events. Probably the main effect of the electron temperature is the sharpness of the onset of the inter-band contribution to the conductivity. This manifests in two ways. On the one hand, it leads to a softer transition between the regimes of low and high optical loss. On the other hand, it impacts the maximally attainable gain figure, because the increased gain at the inter-band threshold and at lower temperatures is to large parts due to a resonant ponderomotive nonlinearity at the inter-band threshold. This means that while the electron gas temperature does not influence the quality factor, it does affect the nonlinear coupling between the optical and plasmonic modes as well as the optical propagation length of the hybrid waveguide.

Given that the electron temperature will have an effect on our results, it is necessary to estimate a realistic temperature in settings such as the examples studied in our manuscript. It appears to us that the topic of hot electrons in ultrafast processes with graphene is not fully understood, yet. We estimate the effect as best we can, a significant degree of uncertainty remains. We use Ref. [8] as a starting point. They report an increase in temperature from room temperature to 1500 K (increase of 1200 K, with plenty of uncertainty) if inter-band transitions are possible. The electron gas temperature is determined by the energy dissipated over one electron-lattice relaxation constant, approximately 1 ps. Since the experiment used femto-second pulses, the entire energy absorbed from one pulse contributed to the electron gas heating. Assuming an absorption of 2.3% (intra-band absorption of a graphene monolayer [8]) from a pulse fluence of 700mJ/m^2 corresponds to an absorbed energy density

of 15 mJ/m^2 in the reference in the inter-band regime. In contrast, we have pulses of at least 10 ps in mind, so we must compare to the product of the relaxation time and the *power* absorbed per unit area of graphene in the waveguide.

In order to get an estimate for our proposal, we assume an optical decay length of 1 mm, which is a realistic number in the low-loss regime $E_F \geq 0.4 \text{ eV}$ at room temperature. Together with a strip width of 150 nm, we find an effective dissipated energy per electron-lattice relaxation time of $\approx 5 \text{ mJ/m}^2$ for each Watt of pump power. The resulting electron heating must be based on the reported 1200 K increase. As a result, we would estimate an electron gas heating in the ballpark of 300 K above the lattice temperature per Watt of pump power. This heating effect should drop by an order of magnitude at 60 K due to reduced optical absorption.

In the manuscript, we project pump powers between 10 mW and 10 W. Clearly, the upper boundary becomes

problematic due to this effect, but it appears that for pulse powers below 300 mW the impact is minor both at room temperature (leading to 100 K of heating) and at 60 K (leading to 10 K heating). Towards the lower end of projected pump power range, we do not anticipate any significant effect.

In order to demonstrate the impact of thermal imbalance, we repeated the calculation with 60 K lattice temperature and 160 K electron gas temperature, which corresponds to 3 W pump power at that temperature and in the low optical loss regime. We find that the absolute gain drops by an order of magnitude and the relative gain by two orders of magnitude, but remains above the detection threshold. It should be noted that silicon also exhibits negative effects with higher powers. For cw excitation the pump power is limited to the order of 10 mW due to the accumulation of free carriers generated by two-photon absorption. This can be remedied by using sub-ns pulses, but it will be very difficult to go beyond peak powers in the Watt range.

-
- [1] C. Wolff, M. J. Steel, B. J. Eggleton, and C. G. Poulton, Phys. Rev. A **92**, 013836 (2015).
- [2] C. Wolff, C. Tserkezis, and N. A. Mortensen, New J. Phys. **21**, 073046 (2019).
- [3] T. Stauber, N. M. R. Peres, and A. K. Geim, Phys. Rev. B **78**, 085432 (2008).
- [4] Y.-C. Chang, C.-H. Liu, C.-H. Liu, S. Zhang, S. R. Marder, E. E. Narimanov, Z. Zhong, and T. B. Norris, Nat. Commun. **7**, 10568 (2016).
- [5] R. W. Boyd, *Nonlinear optics, 3rd edition* (Academic Press, 2003).
- [6] R. Pant, C. G. Poulton, D.-Y. Choi, H. Mcfarlane, S. Hile, E. Li, L. Thevenaz, B. Luther-Davies, S. J. Madden, and B. J. Eggleton, Opt. Express **19**, 8285 (2011).
- [7] R. Van Laer, B. Kuyken, D. Van Thourhout, and R. Baets, Nat. Photon. **9**, 199 (2015).
- [8] G. Soavi, G. Wang, H. Rostami, D. G. Purdie, D. De Fazio, T. Ma, B. Luo, J. Wang, A. K. Ott, D. Yoon, S. A. Bourelle, J. E. Muench, I. Goykhman, S. Dal Conte, M. Celebrano, A. Tomadin, M. Polini, G. Cerullo, and A. C. Ferrari, Nat. Nanotechnol. **13**, 583 (2018).

REVIEWERS' COMMENTS:

Reviewer #3 (Remarks to the Author):

4th Report on the manuscript

"Stimulated plasmon polariton scattering", by C. Wolff and N. A. Mortensen

Following the Editor's request I only comment on the points 4 – 7 of the Authors' Reply.

Point 6. It would be excellent if the authors answered all questions in a similar manner.

Point 5. Ok.

Point 4. The plasmon frequency increases indeed in ribbons as compared to an infinitely extended graphene, but whether the difference can be as large as $\sim 50\%$ remains unclear. This difference depends on the parameter qW where W is the ribbon width, but the authors only add the phrase "We note that the dispersion relation of such ribbons differs from that of infinitely extended graphene due to the lateral confinement" without giving a quantitative estimate. Why do the authors carefully avoid quantitative discussions?

Point 7. This question is not answered at all. The answers of the type "We are the leading group of the world and everybody should believe our statements without a doubt" are unacceptable. We are not in a church to believe; in a theoretical paper every statement should be mathematically proved.

It is well known since ~ 1980 , i.e. 40 years before the publication of the review [33], that the spectrum of 2D plasmons in a gated system is linear at $qd \ll 1$ and that the plasmon frequency in a gated system is much smaller than in an ungated one. Thus, like with the chemical doping (point 3), the use of electrical doping should also lead to a decrease of Q , due to the reduction of the plasmon frequency as compared to the ungated case. Why do the authors avoid a quantitative discussion here again?

Comments of Reviewer 3 (italic font) and our responses

Reviewer:

Following the Editor's request I only comment on the points 4 – 7 of the Authors' Reply.

Point 6. It would be excellent if the authors answered all questions in a similar manner.

Point 5. Ok.

Response:

We are more than happy to discuss in detail anything that might be a fundamental problem to the concept of coherent self-amplifying interaction between THz plasmon polaritons and light in nanoscale waveguides. Electron gas heating is such a case.

Reviewer:

Point 4. The plasmon frequency increases indeed in ribbons as compared to an infinitely extended graphene, but whether the difference can be as large as 50% remains unclear. This difference depends on the parameter qW where W is the ribbon width, but the authors only add the phrase "We note that the dispersion relation of such ribbons differs from that of infinitely extended graphene due to the lateral confinement" without giving a quantitative estimate. Why do the authors carefully avoid quantitative discussions?

Response:

We would like to reiterate that the main claim of our paper is the principle of coherent self-amplifying interaction between THz plasmon polaritons and light in nanoscale waveguides and the example is given to illustrate feasibility. While we took many details such as lateral confinement into account in order to obtain realistic numbers, we feel that a discussion of this (and all the other minor details) would distract from the main message. We did not want to write a design guide. Even worse, a detailed discussion of the design aspects of slot waveguides might create or reinforce the impression that slot waveguides represent the only viable platform. We believe that would be a disservice to the community.

The reviewer is correct: The confinement depends on qW . In our manuscript, q is explicitly stated as twice the optical wave number and W can be seen in Fig. 2 to be 150 nm in our example. Hence, the product qW would be on the order of one (in our example it is 1.7). The reviewer does not have to wonder about the magnitude of the effect: it has already been studied in the literature [34].

Reviewer:

Point 7. This question is not answered at all. The answers of the type “We are the leading group of the world and everybody should believe our statements without a doubt” are unacceptable. We are not in a church to believe; in a theoretical paper every statement should be mathematically proved.

It is well known since 1980, i.e. 40 years before the publication of the review [33], that the spectrum of 2D plasmons in a gated system is linear at $qd \ll 1$ and that the plasmon frequency in a gated system is much smaller than in an ungated one. Thus, like with the chemical doping (point 3), the use of electrical doping should also lead to a decrease of Q , due to the reduction of the plasmon frequency as compared to the ungated case. Why do the authors avoid a quantitative discussion here again?

Response:

We are very sorry if we offended the reviewer, but we can assure them that we have no theological ambition whatsoever. We merely wanted to indicate that we are aware of the effects that metals have on nearby graphene, because we have published research on that very problem.

The reviewer is quite correct that non-quadratic dispersion relations occur for $qd \ll 1$. In our text, we state q as twice the optical wavenumber and propose d on the order of microns. This leads to $qd > 10$. Relevant effects would be expected for $d \ll 100$ nm, which would be unacceptable anyway because of the optical loss introduced by metals.